# Surface–atmosphere exchange of inorganic water–soluble gases and associated ions in bulk aerosol above agricultural grassland pre– and post– fertilisation

Robbie Ramsay[1,2], Chiara F. Di Marco[1], Mathew R. Heal[2], Marsailidh M. Twigg[1], Nicholas Cowan[1], Matthew R. Jones[1], Sarah R. Leeson[1], William J. Bloss[3], Louisa J. Kramer[3], Leigh Crilley[3], Matthias Sörgel[4], Meinrat Andreae[4,5], Eiko Nemitz[1]

[1] Centre for Ecology & Hydrology (CEH), Edinburgh, EH26 0QB, United Kingdom
[2] School of Chemistry, University of Edinburgh, West Mains Road, EH9 3JJ, United Kingdom
[3] School of Geography, Earth and Environmental Sciences, University of Birmingham, Birmingham, B15 2TT, United Kingdom
[4] Max Plank Institute for Chemistry, Biogeochemistry Department, Mainz, P.O. Box 3060, Germany
[5] Scripps Institution of Oceanography, University of California San Diego, La Jolla, CA 92093, USA

Correspondence to: Eiko Nemitz (en@ceh.ac.uk)

**Abstract.** The increasing use of intensive agricultural practices can lead to damaging consequences for the atmosphere through enhanced emissions of air pollutants. However, there are few direct measurements of the surface-atmosphere exchange of trace gases and water–soluble aerosols over agricultural grassland, particularly of reactive nitrogen compounds. In this study, we present measurements of the concentrations, fluxes and deposition velocities of the trace gases HCl, HONO, $HNO_3$, $SO_2$ and $NH_3$, and their associated water-soluble aerosol counterparts $Cl^-$, $NO_2^-$, $NO_3^-$, $SO_4^{2-}$, $NH_4^+$ as determined hourly for one month in May–June 2016 over agricultural grassland near Edinburgh, UK, pre- and post- fertilisation. Measurements were made using the Gradient of Aerosols and Gases Online Registration (GRAEGOR) wet–chemical two–point gradient instrument. Emissions of $NH_3$ peaked at 1460 ng $m^{-2}$ $s^{-1}$ three hours after fertilisation, with an emission of HONO peaking at 4.92 ng $m^{-2}$ $s^{-1}$ occurring five hours after fertilisation. Apparent emissions of $NO_3^-$ aerosol were observed after fertilisation which, coupled with a divergence of $HNO_3$ deposition velocity ($V_d$) from its theoretical maximum value, suggested the reaction of emitted $NH_3$ with atmospheric $HNO_3$ to form ammonium nitrate aerosol. The use of the conservative exchange fluxes of tot-$NH_4^+$ and tot-$NO_3^-$ indicated net emission of tot-$NO_3^-$, implying a ground source of $HNO_3$ after fertilisation. Daytime concentrations of HONO remained above the detection limit (30 ng $m^{-3}$) throughout the campaign, suggesting a daytime source for HONO at the site. Whilst the mean $V_d$ of $NH_4^+$ was with 0.93 mm/s in the range expected for the accumulation mode, the larger average $V_d$ for $Cl^-$ (3.65 mm/s), $NO_3^-$ (1.97 mm/s), $SO_4^{2-}$ (1.89 mm/s) reflected the contribution of a super-micron fraction and decreased with increasing $PM_{2.5}$/$PM_{10}$ ratio (a proxy measurement for aerosol size), providing evidence – although limited by the use of a proxy for aerosol size – of a size-dependence of aerosol deposition velocity for aerosol chemical compounds, which has been suggested from process orientated models of aerosol deposition.

# 1 Introduction

As the demand for food production grows in line with an increasing global population, so too does the development of intensive agricultural practices. These can have deleterious impacts on the environment and human health (Godfray et al., 2010; Foley et al., 2011), particularly through the emission of trace gases and the formation of airborne particles generated by their reactive chemistry. The application of nitrogen-based fertilisers and the keeping of livestock are two systems that are important to the formation of atmospheric reactive nitrogen ($N_r$) compounds, such as the gases ammonia ($NH_3$), and nitrous acid (HONO), the latter of which, together with nitric acid ($HNO_3$), also derives from the oxidation of nitrogen oxides ($NO_x$) emitted by combustion sources. The associated condensed-phase components of ammonium ($NH_4^+$) and nitrate ($NO_3^-$) exist in equilibrium (as ammonium nitrate ($NH_4NO_3$)) with $NH_3$ and $HNO_3$ (Robertson et al., 2013). The emission of these $N_r$ species and their subsequent deposition by washout (wet deposition) or uptake on the surface (dry deposition) have high spatial and temporal variability, and can have critical impacts on terrestrial and aquatic ecosystems, especially those which are nitrogen limited (Galloway et al., 2003; Fowler et al., 2013). The deposition of $NH_3$ has been specifically linked to eutrophication and to changes in ecosystem composition from nitrogen sensitive to nitrogen tolerant plant species (Bobbink et al., 2010), as well as to reduction in biodiversity of coastal waters (Camargo and Alonso, 2006). The seepage of $N_r$ compounds into soil can also affect the nitrification/denitrification cycle, giving rise to increased emissions of the greenhouse gas nitrous oxide ($N_2O$) as well as of nitric oxide (NO), which in turn effects the formation of $HNO_3$ and HONO (Medinets et al., 2014).

As the primary basic gas in the atmosphere, $NH_3$ also reacts with other trace acidic gases, such as hydrogen chloride (HCl) and sulfuric acid ($H_2SO_4$). The products of these reactions give rise to the aerosols, ammonium chloride ($NH_4Cl$) and ammonium sulfate (($NH_4)_2SO_4$), which along with $NH_4NO_3$ act as scattering aerosols that alter the Earth's total albedo and contribute significantly to regional and global climate (Fiore et al., 2015). Ammonium sulfate is particularly long lived, and its transport and subsequent deposition to surfaces such as agricultural soils can affect plant health (van der Eerden et al., 1992) and lower soil pH (Elliott et al., 2008). The dry deposition of the acidic gases themselves can also induce soil acidification, which on agricultural soils can limit the growth of crops through perturbing the uptake of nutrients. The ammonium salts make a significant contribution to inhalable particulate matter (PM) associated with human health impacts, with $NH_4NO_3$ often dominating PM pollution events in northern Europe (Vieno et al., 2014).

It is therefore important that measurements be made of the surface-atmosphere exchange of trace gases and associated aerosol compounds to quantify the emissions from – and deposition to - land used for agriculture. This also provides important process understanding to represent better the dry deposition processes in chemistry and transport models used to predict air quality and climate change. Understanding the impact of agricultural activities on the environment informs the development of abatement strategies and legislation designed to control emissions, for example through instructing agricultural managers on how best to apply their fertiliser inputs.

Measurements of trace gases and associated aerosols are, however, restricted by the availability of appropriate instrumentation, complications in their measurement due to their reactivity and water solubility, as well as the potential interference of gas-particle interactions.

Of particular importance is the interaction between $NH_3$ and $HNO_3$. These gases, and their aerosol equivalents $NH_4^+$ and $NO_3^-$, are the primary contributors to atmospheric $N_r$ dry deposition (Andersen and Hovmand, 1999). The majority of $NH_3$ emissions originate from agricultural sources, either from direct point sources from the application of N-containing fertilisers, or from long-term sources from livestock (Behera et al., 2013). The use of urea as a fertiliser is associated with particularly large losses

of $NH_3$ after application, due to the action of the urease enzyme present in soil, which leads to $NH_3$ volatilization (Suter et al., 2013). Ferm et al. (1998) estimate that fertiliser losses as $NH_3$ average 14% of the N applied. Nitrogen losses from animal waste present on grassland used for sheep grazing has also been observed (Cowan et al., 2015). While $NH_3$ is predominantly deposited close to source, resulting $NH_4^+$ aerosol can be transported over large distances.

$HNO_3$ is primarily formed from the oxidation of nitrogen oxides ($NO_x$), which are principally anthropogenic in origin but also have a soil biogenic origin (Pilegaard, 2013). $HNO_3$ is extremely water soluble and is rapidly removed from the atmosphere through deposition or by gas-particle interactions, leading to a high deposition velocity. The gas-phase equilibrium reaction of $HNO_3$ with $NH_3$, which is dependent upon temperature and relative humidity (Mozurkewich, 1993), gives rise to ammonium nitrate (R1).

$$NH_3 + HNO_3 \rightleftharpoons NH_4NO_3 \hspace{5cm} (R1)$$

Higher temperatures and humidity favour the decomposition of $NH_4NO_3$, and its transportation – and subsequent evaporation – can result in the deposition of reactive nitrogen to the surface. The interaction of $NH_3$ with $HNO_3$ can also lead to over estimation of the $HNO_3$ deposition rate, as the additional sink for $HNO_3$ deposition provided for by the reaction violates the theoretical deposition rate modelled on a zero surface resistance model for $HNO_3$. The dissociation of $NH_4NO_3$ over vegetation

can induce an opposite effect, with apparent emissions of $HNO_3$ occurring with associated high deposition rates for $NO_3^-$ and $NH_4^+$ (Nemitz and Sutton, 2004). The sums of the total ammonium (tot-$NH_4^+$ = $NH_3$ + $NH_4^+$) and of total nitrate (tot-$NO_3^-$ = $HNO_3$ + $NO_3^-$), however, are conservative quantities (Kramm and Dlugi, 1994), and the use of them in the measurement of exchange fluxes can help to account for the $NH_3$-$HNO_3$-$NH_4NO_3$ triad on overall deposition rates.

The reaction is complicated by the presence of $SO_2$ and HCl that also compete with $HNO_3$ for the $NH_3$. HCl, like $HNO_3$, is highly water soluble, is deposited quickly to the surface, and consequently has a high deposition velocity. It can be formed by the reaction of other acidic gases, such as $HNO_3$ and $SO_2$, with sodium chloride found in sea spray (Pio and Harrison, 1983). The reaction of $NH_3$ and HCl gives rise to ammonium chloride, which has a similar volatility as $NH_4NO_3$ (Allen et al., 1989).

SO$_2$, which is the precursor for H$_2$SO$_4$ in the atmosphere, is primarily anthropogenic in origin, being emitted via the burning of fossil fuels that contain sulfur.

HONO is similar to HNO$_3$ in that it can derive from oxidation of NO$_x$ precursors. Although it can be formed homogeneously in the atmosphere by the reaction of the hydroxyl radical OH with NO (R2) (Pagsberg et al., 1997), the rate of this reaction is too slow to account for measured concentrations of HONO. Similarly, the heterogeneous reaction involving the reaction of NO$_2$ with H$_2$O on terrestrial surfaces, while potentially a contributory source to atmospheric HONO, has also been found to be too slow to account for measured concentrations (Kleffman, 2007). HONO is photolized during daytime, being a primary source of OH-radicals depending on the source and sink mechanisms that govern its abundance (Sörgel et al., 2015). However, a growing number of field measurements of non-zero HONO concentrations during the day points to the presence of daytime sources (Acker et al., 2006), including the emissions of HONO from soils (Su et al., 2011; Oswald et al., 2013; Scharko et al., 2015).

$$NO + OH \longrightarrow HONO \qquad\qquad\qquad\qquad (R2)$$

Techniques to measure concentrations and fluxes of these trace gas and associated aerosol components require multispecies quantification, low detection limits and fast temporal resolution. Eddy covariance, the most direct micrometeorological technique for the measurement of trace gas fluxes, requires fast-response sensors that are not available for some species (such as HNO$_3$) or are limited by the time-response and potential for chemical interferences of the inlet (Neuman et al., 1999). While eddy covariance has been used to measure NH$_3$ concentrations using laser absorption spectroscopy, such as through the use of quantum cascade lasers (QCL) (Famulari et al., 2004, Zöll et al., 2016), intercomparisons with more established techniques are still lacking.

The aerodynamic gradient method derives fluxes of a tracer from its vertical concentration gradient, which can be obtained from concentration measurements at two or more heights, avoiding the requirement for fast response measurement. Developments in automated wet chemistry instrumentation have in turn led to the development of the Gradient of Aerosols and Gases Online Registrator (GRAEGOR), a two-point gradient system that measures the concentrations of HCl, HONO, HNO$_3$, SO$_2$ and NH$_3$, and their associated aerosol counterparts Cl$^-$, NO$_2^-$, NO$_3^-$, SO$_4^{2-}$ and NH$_4^+$ (Thomas et al., 2009). One of the advantages of the modified aerodynamic gradient method is the ability to determine the deposition velocities ($V_d$) of chemical tracers, provided the flux and concentration at a reference height have been calculated. With the use of the GRAEGOR, which takes measurements of tracers at two heights over one hour, high-resolution time scale measurements of deposition velocities can be acquired.

Other wet chemistry instruments have also been developed to measure individual species at one height, such as the Long Path Absorption Photometer (LOPAP), which measures concentrations of HONO with fewer artefacts than the GRAEGOR (Heland

et al., 2001). A comparison study between LOPAP HONO measurements and the Gas and Aerosol Collector (GAC) - an instrument which uses similar measurement techniques to the GRAEGOR – was conducted by Dong et al. (2012), but there has not yet been a published comparison between the LOPAP and GRAEGOR in measurements of HONO. Similarly, measurements of trace gases and aerosols above agricultural grassland using the GRAEGOR are limited, and previous studies above these land systems have been restricted to measurements of a limited number of species within a limited particle size range.

The aim of this study was to use the GRAEGOR to measure concentrations and fluxes of the trace gases HCl, HONO, $HNO_3$, $SO_2$ and $NH_3$ and their water-soluble aerosol counterparts $Cl^-$, $NO_2^-$, $NO_3^-$, $SO_4^{2-}$ and $NH_4^+$ over agricultural grassland in Scotland during a period in early summer (May–June 2016) that included a fertilisation event using urea pellets. The possible formation of $NH_4NO_3$ post fertilisation, a link between aerosol deposition velocity and size (specifically, a proxy for size based on the $PM_{2.5}/PM_{10}$ ratio from measurements nearby), and the potential ground source formation of HONO are discussed. A further aim of this study was to undertake intercomparisons between the measurements of HONO by the GRAEGOR and two LOPAP instruments, and between measurements of $NH_3$ recorded by a parallel quantum cascade laser eddy covariance system.

## 2 Methodology

### 2.1 Easter Bush Site Description

The campaign was conducted during the late spring/summer 2016 (21st May – 24th June) at the Easter Bush measurement site (3°12'W, 55°52'N, 190 m above sea level), located 10 km south of Edinburgh, UK. Measurements were made at a 3 m tower situated on the boundary of two intensively managed grassland fields (hereafter referred to as North and South Field) of 16 ha total area, composed principally of *Lolium perenne* (perennial Rye grass) (Fig. 1). Due to the presence of the Pentland Hills close by to the west, local wind direction is channelled such that SW winds – the predominant wind direction at the site – yield flux footprints over the South field, while NE winds produces flux footprints over the North field.

Both fields are used for year-round (although not continuous) sheep grazing, in rotation with adjacent fields, but the South Field also typically has an annual cutting for silage. Mineral fertilisation is carried out twice a year on both fields. During this study, fertilisation of the two fields occurred between 08:00 – 09:00 on the 13th June, using urea mineral fertiliser at a rate of 69.9 kg N ha$^{-1}$. In preparation for this application, sheep that had been present in the fields since April were removed from the South Field on the 2nd June and removed from the North Field on the 9th June. Sheep were reintroduced to the North Field on the 21st June.

Over the years the Easter Bush field site has hosted several long-term measurements of $CO_2$, $CH_4$ and $NO_2$, and has participated in a number of international projects, such as GRAMINAE (GRassland AMmonia INteractions Across Europe) (Sutton et al.,

2009), Greengrass (Soussana et al., 2007) and NitroEurope (Sutton et al., 2007). It has also supported several individual campaigns of trace gas measurements (Di Marco et al., 2004; Famulari et al., 2004; Jones et al., 2017). In particular, fluxes of $NH_3$ were measured over an 18-month period (Milford et al., 2000) and the GRAEGOR was operated during a period of manure application (Twigg et al., 2011).

## 2.2 Instrumentation

### 2.2.1 Gradient of Aerosols and Gases Online Registrator

The GRAEGOR (Energy Research Centre of the Netherlands) is a wet chemistry instrument that measures the concentrations of reactive trace gases (HCl, HONO, $HNO_3$, $SO_2$ and $NH_3$) and water-soluble aerosols ($Cl^-$, $NO_2^-$, $NO_3^-$, $SO_4^{2-}$, $NH_4^+$) continuously, semi-autonomously, and with online analysis at hourly resolution (Thomas et al., 2009; Wolff et al., 2010). The instrument consists of two sampling boxes placed at two heights (during this campaign, $z_1 = 0.6$ m, $z_2 = 2.4$ m), from which concentration gradients and hence fluxes can be derived.

Each sample box contains a horizontal wet rotating annular denuder (WRD) (Keuken, 1988) and a steam jet aerosol collector (SJAC) (Khylstov et al., 1995; Slanina et al., 2001) connected in series. Air is drawn through each sample box simultaneously by an air pump at a rate of 16.7 L min$^{-1}$, passing first through the WRD, which is continuously coated with a feeding solution of double-deionized water (DDI) of 18.2 MΩ resistance. Trace gases within the laminar air flow are absorbed into the sorption solution which is then fed from the sample box to a detection unit located at ground level. The trace-gas-free air then passes through the SJAC, where particles within the air flow are mixed with steam generated from the DDI water feeding solution, precipitating a supersaturation event causing the water–soluble particles to grow into droplets. The enlarged droplets are separated out of the air stream by a cyclone and fed as a liquid sample to the detection unit. Liquid samples from the SJAC and WRD of each sample box are analysed for $NH_3/NH_4^+$ using flow injection analysis (FIA) (Wyers, 1993; Norman et al., 2009). An ion chromatography (IC) unit equipped with a Dionex AS12 column, quantifies the concentration of $HONO/NO_2^-$, $HNO_3/NO_3^-$ and $SO_2/SO_4^{2-}$ based on the measured conductivity of the respective anions within the liquid sample compared to a reference standard of 50 ppb $Br^-$ added to the sample solution. Analysis by FIA and IC is carried out over 15 minutes, and using a flow control scheme, a half-hourly averaged concentration of trace gases and water-soluble aerosols is generated for each height every hour.

A high-density polyethylene tube (0.3 m length, and 1/3" outer diameter) with a HDPE filter is placed at the inlet of the WRD in order to minimise the loss of $HNO_3$ and $NH_3$ and to ensure a particle diameter cutoff of 0.2 nm. A biocide of 0.6 mL of hydrogen peroxide (30%) is added to every 1 L of the DDI water feeding solution to prevent biological contamination in the WRD of each sample box. Air flow is controlled using a critical orifice downstream of the SJAC.

Autonomous calibration of the FIA system was carried out 24 h after the beginning of the campaign, and every 72 h thereafter, giving a total of 5 internal calibrations of this system. Calibration was conducted using three liquid $NH_4^+$ standards of 0, 50 and 500 ppb concentration. The IC unit is continuously checked for analytical performance by the addition of a liquid $Br^-$ internal standard (50 ppb concentration) to each column injection. Calibration of the IC unit was conducted twice during the
campaign (23[rd] May and the 28[th] June, prior to and after the campaign respectively) using a mixed ionic liquid standard consisting of 25 ppb $SO_4^{2-}$, 20 ppb $NO_3^-$ and 20 ppb $Cl^-$.

Measurements of the air flow into the sample boxes were conducted using an independent device (TSI Mass Flowmeter 4140) once every fortnight during the campaign. Additional checks of the field performance of the instrument included daily checks
of the WRD tubes and sample box air inlets for signs of visible contamination.

The GRAEGOR sampling boxes have very short inlets with no size-selection. Consequently, the aerosol concentration reflects water-soluble total suspended particulate (TSP). It detects any compound that dissociates to form the measured ions and therefore has a number of artefacts. These include interferences in HONO measurements through $NO_2$, particularly during periods
of high $SO_2$ concentrations (Spindler et al., 2003); the inclusion of dinitrogen pentoxide ($N_2O_5$) concentrations in measurements of $HNO_3$ during the night-time measurement periods, though the magnitude of this unclear in rural environments (Phillips et al., 2013); and the potential for organic chloride compounds to be included in measurements of overall $Cl^-$ aerosol (Nemitz et al., 2000a).

The GRAEGOR has been demonstrated to be capable of measuring fluxes in a number of studies both in identical form to the one used here (Wolff et al., 2010, 2011; Twigg et al., 2011) or in related variants (Nemitz et al, 2004; Rumsey and Walker, 2016). Ammonia-specific instruments based on the same technology (AMANDA, GRAHAM, ECN Petten, NL; Wyers et al., 1993) represent the most commonly used instrument for the automated measurement of ammonia fluxes.

### 2.2.2 Supplementary Measurements

Vertical profiles of temperature were measured at the tower using fine-thread, custom-made thermocouples set at the same heights as the GRAEGOR sample boxes. Located 0.4 m from the tower, an eddy covariance system (Gill Anemometer R01012 with LI-COR-7000) at a height of 2.6 m measured three-dimensional wind speed, sensible heat flux ($H$), frictional velocity ($u_*$) and wind direction. Ongoing, long-term measurements of relative humidity (RH) (Vaisla 50/Y humitter), global radiation (Skye Instruments SKS 110 pyranometer), and total rainfall (Campbell Scientific ARG110 tipping bucket rain gauge) were
also available at the site for the campaign period. Measurements of HONO taken by a LOPAP (QUMA Electronik &Analytik, Wuppertal, Germany) and $NH_3$ measurements taken by a Quantum Cascade laser (QCL) (Aerodyne Research Inc., Billerica, USA) during the campaign period were used for comparison studies with GRAEGOR measurements. Measurements of $NO_2$

concertation, used in Section 4.4.1 to quantify an artefact in GRAGOR HONO measurements, were recorded by a chemilumi-nescence $NO_2$ detector (200E, Teledyne API, San Diego, California, USA) located 300 m south-east of the Easter Bush site.

## 2.3 Micrometeorological Theory

### 2.3.1 Aerodynamic Gradient Method

The aerodynamic gradient method (AGM), based upon flux-gradient similarity theory, calculates the flux of a tracer ($\chi$, such as a gas or aerosol species) based on its vertical concentration gradient coupled with turbulence parameters (Fokken, 2008). In this paper a hybrid version of the AGM is used, in which the flux is calculated as (Flechard, 1998):

$$F_\chi = -u * \kappa \frac{\partial \chi}{ln\left(\frac{z_2 - d}{z_1 - d}\right) - \psi_H\left(\frac{z_2 - d}{L}\right) + \psi_H\left(\frac{z_1 - d}{L}\right)} \tag{1}$$

where the friction velocity ($u_*$) is derived from eddy-covariance measurements with a sonic anemometer; $\kappa$ is the von Karman
constant ($\kappa = 0.41$); $z_2$ and $z_1$ are the heights of the sample boxes; $d$ is the displacement height; and $\zeta$ is a dimensionless stability parameter expressing the ratio $(z-d)/L$, where $L$ is the Obukhov length, a measure for atmospheric stability. The parameter $\Psi_H$, an integrated form of the heat stability correction term, accounts for deviations from the log-linear profile under non-neutral stratification. By convention, negative and positive flux values denote deposition and emission, respectively.

### 2.3.2 Choice of displacement height, *d*, value

A temperature gradient profile for the campaign was derived from measurements of air temperature at the two heights at which concentrations were measured (0.6 m and 2.4 m). Sensible heat flux (*H*) was calculated from the temperature gradient as per Wang and Bras (1998):

$$H = -\rho_{air} \, c_p \, K_H(z - d)\frac{dT}{dz}, \tag{2}$$

where $c_p$ is the heat capacity of air, $\rho_{air}$ is the density of air, and $K_H$ is the eddy diffusivity constant for heat. $K_H$ can be calculated
as

$$K_H(z - d) = \frac{(z-d)u_*}{\phi_H\left(\frac{z-d}{L}\right)}, \tag{3}$$

where $z$ is the absolute height above ground, $d$ is the displacement height, $u_*$ is the friction velocity, and $\Phi_H$ is the stability correction for sensible heat. Sensible heat flux and, by extension, the flux of the trace gas and aerosol species, are dependent upon the value of $d$. In order to ensure that the correct displacement height was chosen, the sensible heat flux based upon the
temperature gradient developed from thermocouple measurements was calculated using a variety of different values for dis-placement height. The resulting values for the sensible heat flux were then compared through linear regression to the value for the sensible heat flux recorded by the eddy covariance system also present. A displacement height value of 0.14 m gave the

closest agreement between the sensible heat fluxes derived by the aerodynamic gradient approach and eddy-covariance, with a linear regression slope of 0.997 and $R^2 = 0.945$.

### 2.3.3 Determination of Dry Deposition Velocities

The dry deposition velocity ($V_d$) of a tracer is the negative ratio of its flux to its concentration ($\chi$) at height z – d

$$V_d(z-d) = -\frac{F_\chi}{\chi_z(z-d)}, \tag{4}$$

The $V_d$ for gas species may also be expressed as the reciprocal of the total resistance for deposition, which is composed of $R_a$ (the aerodynamic resistance), $R_b$ (the quasi-laminar boundary layer resistance) and $R_c$ (the canopy resistance) as per the resistance analogy for dry deposition (Fowler and Unsworth, 1979; Wesley, 1989). $R_a$ and $R_b$ were calculated from (5) and (6) using meteorological measurements taken at the site using (Garland, 1977)

$$R_a(z-d) = \frac{u(z-d)}{u_*^2} - \frac{\psi_H(\zeta) - \psi_M(\zeta)}{\kappa u_*}, \tag{5}$$

$$R_b = (Bu_*)^{-1}, \tag{6}$$

where $B^{-1}$, B being the Stanton number, is parametrized by the turbulent Reynold's number, $R_{e*}$ (the ratio of the frictional force to the kinematic velocity of air) and the Schmidt number, $S_e$, (the ratio of kinematic velocity of air to the molecular diffusivity coefficient of the gas species) :

$$B^{-1} = 1.45 R_{e_*}^{0.24} S_c^{0.8}, \tag{7}$$

If $R_a$ and $R_b$ are calculated from measurements, $R_c$ can be inferred via:

$$R_c = \frac{1}{V_d(z-d)} - R_a(z-d) - R_b, \tag{8}$$

For gases, a theoretical maximum deposition velocity can be calculated when it is assumed that the gas is completely absorbed by the canopy (i.e. for $R_c = 0$):

$$V_{max} = \frac{1}{(R_a + R_b)}, \tag{9}$$

The canopy resistance approach can only describe deposition and fails when the exchange of a gas is bi-directional, such as often the case with $NH_3$. In this case, the canopy compensation point model can be adopted, which considers the surface interaction of $NH_3$ in terms of parallel resistance pathways, composed of individual resistances such as stomatal resistance and cuticular resistance (Nemitz et al., 2000b; Flechard et al., 1999).

The gradient technique is only applicable for inert species whose flux is constant with height. Most studies of surface-exchange fluxes of reactive compounds do not have the information to assess whether chemical reactions might interfere with the flux

measurement, but in this study the behaviour of HNO$_3$ and HCl allows us to draw conclusions on flux divergence (Section 4.2.1). Following precedence in the literature (e.g. Nemitz et al., 2004) we initially evaluate fluxes assuming that chemistry can be ignored, and then discuss the validity of this discussion based on the results.

### 2.3.4 Limits of detection and estimation of uncertainties in concentration measurements and flux calculations

The concentration limit of detection (LOD) of the instrument for each of the species measured was quantified from a field blank test. The field blank test was carried out prior to the campaign on the 20[th] March over 24 hours by switching off the sample box air pump and sealing the air inlets, but leaving the rest of the system unaltered, as per Thomas (2009). Limits of detection were then calculated as three standard deviations from the average background signal. Results from this test are presented in Table 1, expressed as LOD values for each trace gas and corresponding water–soluble aerosol species.

The minimum detectable flux for each aerosol and gas species measured by the GRAEGOR is dependent upon atmospheric stability and the ambient concentration of the given trace gas or aerosol species. Based on the method described by Thomas et al. (2009), median minimum detectable fluxes ($F_{LOD}$) were calculated for each trace gas and aerosol species measured and are detailed in Tables 2 and 3 respectively.

When calculating the flux of a species using the aerodynamic gradient method, it is apparent that errors in individual concentration measurements propagate into an error in the concentration differences, and subsequently, affect the accuracy of the calculated vertical concentration gradient. Some errors systematically affect both heights and therefore affect the gradient to a lesser extent than systemic errors in sampling efficiency at a single height, such as difference in capture efficiency of the WRD

tubes or slight differences in air flow caused by differences in the critical orifices, may impact on the accuracy of concentration measurements and resultantly affect the precision in the error of concentration difference.

The overall random error in the measurements of the trace gas and water–soluble aerosol concentrations ($\sigma_m$) can be determined using a Gaussian error propagation approach, in which the concentration error is expressed as a product of several individual

measurement errors with the mixing ratio, $m$ (Trebs et al., 2004) (eq. 9) –

$$\sigma_m = m \sqrt{(\frac{\sigma_{m_{liq}}}{m_{liq}})^2 + (\frac{\sigma_{Br(std)}}{Br_{(std)}})^2 + (\frac{\sigma_{Q_{Br}}}{Q_{Br}})^2 + (\frac{\sigma_{m_{Br}}}{m_{Br}})^2 + (\frac{\sigma_{Q_{air}}}{Q_{air}})^2}$$ (10)

Here, $m_{liq}$ is the mixing ratio of the compounds found in the analysed liquid sample in ppb, $Br_{(std)}$ the stated mixing ratio of the

internal Br$^-$ standard, $Q_{Br}$ the flow rate of the internal Br$^-$ standard, $m_{Br}$ the analysed Br$^-$ mixing ratio and $Q_{air}$ the air mass flow through the system. All values have an associated standard deviation, $\sigma_x$. This formulation holds strictly for the species measured by ion chromatography; for NH$_3$ and NH$_4^+$, the equation is altered by omitting the factor relating to Br$^-$ addition and

substituting the factor for $Q_{Br}$ and its associated standard deviation with the term $Q_S$, the flow of the analysed liquid sample of $NH_3$ or $NH_4^+$.

Uncertainties for the trace gases and water–soluble aerosols measured calculated by error propagation ranged from 8% - 18%

($3\sigma$) throughout the campaign, varying primarily due to fluctuations in the measured flow rate and analysed concentration of the internal $Br^-$ standard.

The error in the concentration difference ($\sigma_{\Delta c}$) can be characterised experimentally, by placing both sample boxes can at one height, or – provided that the absolute difference between sample heights is small – by using one common air inlet at a specified

height, with the instrument operated normally. From this side-by-side measurement, linear regression analysis accompanied by orthogonal best of fit between the concentrations measured by each sample box can be conducted, with deviation from a 1:1 fit between sample heights defined as a systemic error. Using the calculated orthogonal fit equation, corrections in the concentrations can then be applied, accounting for the systemic bias (Wolff et al., 2010b). After correction using the orthogonal fit, the remaining scatter – termed the residuals – was used to determine the error in the concentration difference. During this

campaign, one side-by-side measurement was conducted on the 8$^{th}$ June for 16 hours by connecting a common air inlet set at $z = 1.2$ m between each sample box. From the results obtained, it was found that for the gases $NH_3$, HCl, HONO, $HNO_3$ and $SO_2$ that deviation from the 1:1 fit resulted in a precision of measurements <4% ($3\sigma$). For the aerosol species $Cl^-$, $NO_3^-$ and $SO_4^{2-}$, precision was calculated as <8% ($3\sigma$), while for $NH_4^+$ was calculated as <9% ($3\sigma$).

Errors in flux calculations can similarly be determined through the Gaussian error propagation method applied to Eq. (1). Wolff et al. (2010b), using an analogous form of this equation, showed that total error in the flux is composed of ($\sigma_{\Delta c}$) and the error in the flux-gradient relationship (expressed as a transport velocity by Wolff et al., 2010b), which is dominated by the error in $u_*$ ($\sigma_{u*}$).

$$\sigma_F = F \sqrt{(\frac{\sigma_{vu_*}}{u_*})^2 + (\frac{\sigma_{\Delta c}}{\Delta_c})^2} \tag{11}$$

This simplification neglects the detailed secondary errors associated with the stability correction which to quantify fully is beyond the scope of this paper.

$\sigma_{u*}$ is dependent upon the sonic anemometer used and whether conditions are neutral or non-neutral (Foken, 2008; Nemitz et al., 2009a). For neutral conditions, and based on the sonic anemometer used, $\sigma_{u*}$ was estimated at $\leq$ 10%. For non-neutral conditions, $\sigma_{u*}$ was estimated at 12% median, which, in combination with $\sigma_{\Delta c}$, was used to calculated $\sigma_F$.

Throughout this paper, stated errors for concentration measurements are derived from the measurement uncertainty as calculated by Eq. (10), while stated errors for flux calculations are derived from the flux uncertainty as calculated by Eq. (11). Calculated errors for the uncertainty in concentration measurements, the error in the concentration difference, and the error in the calculated fluxes for all species measured are similar to values determined by previous studies which have used the GRAEGOR successfully to measure flux gradients (Thomas et al., 2009; Wolff et al., 2010; Twigg et al., 2011).

### 2.3.5 Data Post Processing

Concentrations that were less than five times the limit of detection as calculated before the campaign began (20[th] March) were discarded. Calculated fluxes were filtered according to a standard protocol. Fluxes were not calculated for periods of low wind speed ($u < 1$ m s$^{-1}$), low friction velocity ($u < 0.15$ m s$^{-1}$), and very stable conditions as indicated by the Obukhov length absolute value ($|L| < 5$ m). Fluxes were also discarded for periods when the wind was obstructed by the measurement cabin and other towers (270°> wd < 320°, and 120°> wd <160°). Calculated fluxes which were below the minimum detectable flux value for their respective trace gas or aerosol species were discarded.

## 3 Results

### 3.1 Meteorology

Figure 2 shows time series of the rainfall, radiation, relative humidity, air temperature and wind speed and direction measured during the campaign. The meteorology splits into two episodes. From 24[th] May to 5[th] June 2016, the dominant prevailing wind direction was north easterly, accompanied by dry and sunny conditions with air temperature displaying a characteristic diel cycle that increased each day. Following a period of cloudier conditions from 6[th] to 10[th] June, the prevailing wind direction shifted to south westerly for the remainder of the measurement period. Conditions became wetter and the diel air temperature amplitude was reduced. Relative humidity remained high throughout the campaign, with only occasional periods <70%, such as 3[rd] – 4[th] June and the 21[st] – 23[rd] June. Wind speed was variable throughout, ranging between 0.05 and 5.87 m s$^{-1}$, with a median value of 2.16 m s$^{-1}$. During the fertilisation period, the prevailing wind direction was from the SW, and therefore over the South Field, with no precipitation but high (>90%) relative humidity.

### 3.2 Concentrations of trace gases and water-soluble aerosols

Summary statistics for the concentrations of the trace gas and water-soluble aerosol species measured at 2.4 m during the campaign are presented in Table 1. Median values for the concentrations of water-soluble aerosol species were similar to those measured in PM$_{10}$ at the nearby rural background monitoring site of Auchencorth Moss (Twigg et al., 2015). The time series of the measured aerosol and trace gas concentrations are displayed in Figures 3 and 4, respectively. Data gaps in the time series are due to in-field calibrations, poor chromatograms, or instability in liquid or air flow.

Mean concentrations of $NO_3^-$ were 1.53 µg m$^{-3}$ (2.4 m), whereas its gaseous counterpart, $HNO_3$, had mean concentrations of 0.19 µg m$^{-3}$ (2.4 m). The mean particulate $NO_3^-$ concentrations were therefore almost 6 times greater than the gaseous $HNO_3$ counterpart. The same dominance of particulate $SO_4^{2-}$ concentrations over gaseous $SO_2$ concentrations was also observed.

5 Median concentrations of particulate $Cl^-$ were 0.37 µg m$^{-3}$ and 0.36 µg m$^{-3}$ at 0.6 m and 2.4 m, respectively. The mean concentrations of $Cl^-$ were also similar at both heights at 0.89 µg m$^{-3}$ and 0.91 µg m$^{-3}$, respectively. Variation in HCl concentrations at each height was more pronounced, with a mean value of 0.16 µg m$^{-3}$ at 0.6 and 0.20 µg m$^{-3}$ at 2.4 m, and a median value of 0.12 µg m$^{-3}$ at 0.6 m and 0.15 µg m$^{-3}$ at 2.4 m. As for particulate $NO_3^-$ and gaseous $HNO_3$, measured particulate $Cl^-$ concentrations were greater than those of gaseous HCl, by about a factor of 2 at each height.

In contrast, $NH_3$ concentrations were larger than those of particulate $NH_4^+$; median concentrations of $NH_3$ were 1.15 µg m$^{-3}$ (2.4 m), while median concentrations of $NH_4^+$ were 0.64 µg m$^{-3}$ (2.4 m). The average concentrations of $NH_3$ were similar to those reported previously at the same site for the same time of year (Milford, 2004). Similarly, the average concentrations of HONO are higher than those of its particulate counterpart, $NO_2^-$, with median concentrations for HONO of 0.04 µg m$^{-3}$ (2.4 m) and corresponding concentrations for $NO_2^-$ 0.02 µg m$^{-3}$ (2.4 m) respectively.

Maximum concentrations for $NH_3$ and HONO at 0.6 m were 21.4 µg m$^{-3}$ and 0.15 µg m$^{-3}$. At 2.4 m, the maximum concentration for $NH_3$ and HONO were 13.8 µg m$^{-3}$ and 0.12 µg m$^{-3}$. The maximum values at each height occurred at 11:00 on the 13[th] June for $NH_3$, one hour after fertilisation of the South Field, and at 13:00 on the 13[th] June for HONO, four hours after fertilisation of the South Field.

The time series of measurements presented in Figures 3 and 4 show that both aerosol and trace gas concentrations are affected by prevailing meteorological conditions, with larger concentrations for each species during the drier, warmer period of 28[th] May to 6[th] June, followed by decreased concentrations from 6[th] to 10[th] June when precipitation increased and temperature decreased. Concentrations were lower – except for the peaks in $NH_3$ and HONO after fertilisation on the 13[th] June - during the period from 10[th] June to the end of the campaign, concurrent with the change in prevailing wind direction from the NE to the SW.

The concentrations of $HNO_3$ and $SO_2$ showed a strong diel cycle (Figure 4) from the 26[th] May to the 9[th] June, with maxima at both measurement heights occurring between 11:00 and 14:00, and minima occurring at night between 03:00 and 06:00. A similar, but weaker, inverted pattern was exhibited by their particulate counterparts, with $NO_3^-$ concentrations at both heights (Figure 3) having maxima between 02:00 and 04:00, and minima between 12:00 and 15:00.

Figure 5 shows the median diel concentrations of $NH_3$, $HCl$, $HONO$, $HNO_3$ and $SO_2$ at 2.4 m prior to fertilisation. The median concentrations of HONO remained above the detection limits of the instrument even during daytime, contrary to its expected photochemistry. While concentrations of HONO peaked during night-time and decreased during the day as incoming solar radiation increases, there remained a detectable concentration of HONO at both heights even for the measurement minima at 15:00. The median diel concentrations for $HCl$, $HNO_3$ and $SO_2$ show a shared pattern, with concentrations peaking during the day to reach a maximum between 11:00 to 14:00, followed by a decrease during the night, reaching minima between 02:00 and 04:00. The concentrations of $NH_3$ showed little variation across the day. Figure 6 shows the median diel concentrations of $NH_4^+$, $Cl^-$, $NO_3^-$ and $SO_4^{2-}$ at 2.4 m prior to fertilisation. The median diel concentrations of $NH_4^+$ reach a minimum at 16:00, with a maximum at 02:00. The concentrations of $NO_3^-$ show a similar pattern of early morning median maxima (04:00) and afternoon minima (13:00). The median diel $SO_4^{2-}$ concentrations had maxima at midnight and a minimum at 16:00. The $Cl^-$ concentrations reached a maximum at 03:00 and a minimum at 13:00; however, the upper quartile range was high across all hours, with the maximum concentration of 7.88 µg m$^{-3}$ recorded at 03:00 (median at this time is 0.5 µg m$^{-3}$).

### 3.3 Fluxes, Deposition Velocities, and Canopy Resistance

### 3.3.1 Fluxes of trace gases

Figure 7 shows the time series of the fluxes for the traces gases measured during the campaign. Data gaps are due to either absent data points (unpaired concentrations), or periods where data were filtered (refer to Section 2.3.4).

Bi-directional fluxes were present for both $NH_3$ and HONO, with emission events for each gas occurring during the period of fertilisation of the South Field. For the other trace gases – $HCl$, $HNO_3$ and $SO_2$ – the flux was uni-directional, with deposition occurring throughout the campaign. The deposition for $HCl$, $HNO_3$ and $SO_2$ varied, with larger deposition fluxes occurring during the warmer, drier periods, particularly during the period 1$^{st}$ - 8$^{th}$ June, and smaller deposition fluxes close to zero during the colder, wetter period at the end of the campaign (15$^{th}$ - 24$^{th}$ June).

Summary statistics for the trace gas fluxes, deposition velocities, theoretical maximum deposition velocities, and canopy re-sistance values are presented in Table 2. The maximum $NH_3$ flux was +1460 ng m$^{-2}$ s$^{-1}$, recorded at 12:00 on the 13$^{th}$ June, three hours after fertilisation. The mean flux for $NH_3$ was +15.24 ng m$^{-2}$ s$^{-1}$, suggesting that emission was the predominant flux for $NH_3$ during this campaign. For all other gases, the mean flux values were negative, suggesting that deposition was the net flux process overall. However, a maximum flux for HONO of +4.92 ng m$^{-2}$ s$^{-1}$, recorded five hours after fertilisation, highlights the bi-directional flux pattern for HONO during the campaign. The maximum HONO flux measured here was particularly large. Nitrous acid emissions have previously been reported post fertilisation of grassland using cattle slurry at the same field site ranging from +1.0 to +1.5 ng m$^{-2}$ s$^{-1}$ (Twigg et al. 2011). Table 2 also shows the median relative flux error, the typical flux detection limits ($F_{LOD}$) and the fraction of 60-minute flux values that exceed the $F_{LOD}$ of that period, based on

actual concentration and turbulence. It should be noted that the uncertainty of the campaign averages is much smaller as random uncertainty reduces with the square root of the number of observation that enter the calculation of the ensemble average (e.g. Langford et al., 2015).

Median diel cycles for the deposition velocity and calculated theoretical maximum deposition velocity for the trace gases HCl, HONO, $HNO_3$ and $SO_2$ are shown in Figure 8. The calculation of median diel values for trace gas deposition velocities and canopy resistances excludes the period of flux divergence which occurred during fertilisation. The diurnal deposition velocities for HCl and $HNO_3$ were very close to the calculated maximum deposition velocities, which is expected as a result of their reactivity and high water solubility. The deposition velocity for $SO_2$ is near the theoretical maximum during night–time but is

lower during daytime. The deposition velocity for HONO was consistently lower than its theoretical maximum throughout the entire day. While median values for the $V_d$ for $HNO_3$ are close to the values for $V_{max}$, deposition velocities were recorded that exceeded their corresponding theoretical maximum. While most exceedances fall within the uncertainty range of the measurement, a maximum deposition velocity of 56.8 mm s$^{-1}$ was recorded at 14:00 on the 13[th] June, four hours after fertilisation.

### 3.3.2 Fluxes of water-soluble aerosol components

The measured surface fluxes of the aerosol species Cl$^-$, $NO_3^-$, $SO_4^{2-}$ and $NH_4^+$ are shown in Figure 9, as well as the summary statistics for the fluxes and deposition velocities in Table 3. A large data gap in $NH_4^+$ fluxes from 31[st] May to 10[th] June 2016 was due to $NH_4^+$ only being measured at one height on account of unreliable data for $NH_4^+$ at the lower height of 0.6 m.

Pre-fertilisation, all aerosol species exhibited deposition fluxes. The deposition fluxes were larger during the drier, warmer

period from 31[st] May to 6[th] June, and close to zero during the wetter conditions at the end of the campaign. An important exception was the emission of $NH_4^+$ and $NO_3^-$ from 13:00 on the 13[th] June to 02:00 on the 14[th] June, starting 4 hours after fertilisation of the South Field.

Summary statistics for the fluxes and deposition velocities for the aerosol species measured are shown in Table 3. As for the

trace gases, the median deposition velocities for the aerosol species excludes the period of flux divergence which occurred during fertilisation. The maximum flux for $NH_4^+$ of +18.16 ng m$^{-2}$ s$^{-1}$ was recorded at 16:00 on the 13[th] June, seven hours after fertilisation of the South Field. Similarly, the maximum flux for $NO_3^-$ (+31.84 ng m$^{-2}$ s$^{-1}$) was also recorded soon after fertilisation, at 18:00 on the 13[th] June. Overall, however, the mean fluxes for all aerosol species were negative, confirming a predominant net deposition to the surface.

### 3.4 HONO and NH₃ GRAEGOR measurement comparisons with LOPAP and QCL

### 3.4.1 HONO Comparison Study between GRAEGOR and LOPAP

A comparison of HONO measurements from the GRAEGOR and two LOPAP instruments was conducted from the 26th May to the 6th June to investigate the potential artefacts in the WRD method used by the GRAEGOR. The LOPAPs were part of a study to investigate the mechanisms controlling HONO fluxes over managed grassland, including investigating the potential ground sources of HONO, details of which are presented in Di Marco *et al.* (in preparation). A series of simple linear regression analyses was conducted to determine the level of agreement between the concentrations of HONO measured by each sample box of the GRAEGOR and each of the LOPAPs. The two LOPAP instruments were operated at the two heights of 0.6 m and 2.0 m (hereafter referred to as LOPAP (0.6 m) and LOPAP (2.0m) respectively). In all comparisons, the GRAEGOR recorded a higher concentration of HONO than either of the LOPAPs. The linear regressions suggest that there is a consistent offset in all GRAEGOR concentrations, varying between 0.01 $\mu$g m$^{-3}$ and 0.02 $\mu$g m$^{-3}$. In comparisons between the GRAEGOR Sample Box 1 at 0.6 m and both LOPAPs, the linear concentration relation for HONO varies from 0.92 to 0.97. The comparisons between the GRAEGOR Sample Box 2 (2.4 m) and the LOPAPs suggest that the linear concentration relation for HONO is 1.06 and 1.01 for LOPAP (2.0 m) and LOPAP (0.6m) respectively. In all comparisons, however, there exists a constant concentration offset, which results in a constant higher concentration recorded by both GRAEGOR sample boxes. The closest agreement is between GRAEGOR Sample Box 2 (set at height 2.4 m) and LOPAP (2.0 m), where the HONO concentration recorded by the GRAEGOR Sample Box 2 (2.4 m) is 1.06 that of LOPAP(2.0 m). This comparison also has the best statistical agreement, with an $R^2$ value of 0.67, suggesting a reasonable agreement between the GRAEGOR Sample Box 2 and LOPAP (2.0 m) measurements.

### 3.4.2 NH₃ Comparison Study between GRAEGOR and QCL

On the 7th June, a QCL with inlet at height 1.6 m was installed at the Easter Bush site and took measurements of NH₃ from 19th June to 7th August. Three days of concurrent NH₃ measurements taken by the GRAEGOR and the QCL were recorded in the period 21st – 24th June. The time series of the NH₃ measurements by each instrument are shown in Figure 10(a). An averaged NH₃ concentration at 1.0 m ($\chi$ (1 m)) taken by the GRAEGOR was compared with the NH₃ concentrations taken by the QCL in a simple linear regression analysis, displayed in Figure 10(b). The linear regression shows that the GRAEGOR recorded a factor 1.22 higher concentrations of NH₃ than the QCL, with an associated $R^2$ value of 0.76. However, the number of concurrent measurements is small, with only 41 shared hourly measurement values across three days and a period of 19 continuous hours missing between 02:00 and 23:00 of the 23rd June.

## 4 Discussion

### 4.1 Ion Balance

The ion balance for the hourly-measured cation ($NH_4^+$) and anion ($NO_3^-$ and $SO_4^{2-}$) aerosol species pre-fertilisation is shown in Figure 11. Values are shown as molar equivalent concentration, derived from aerosol mass concentrations converted to

molar concentrations and subsequently multiplied by their charge. $Cl^-$ charge was not included, under the assumption that it would be entirely associated, in the form of sea salt, with $Na^+$ which was not measured by the GRAEGOR. While the correlation between cation and anion species is very good ($R^2 = 0.71$), the linear regression suggests a deficit of $NH_4^+$, suggesting that some of the $NO_3^-$ and/or $SO_4^{2-}$ was balanced by ions other than $NH_4^+$. A likely candidate is $Na^+$: some of the $SO_4^{2-}$ is likely to have represented sea-salt $SO_4^{2-}$ and some $NaNO_3$ is formed by reaction of $NaCl$ with $HNO_3$. Figure 11 is coloured by $Cl^-$

concentration, and periods of anion excess tend to be associated with elevated $Cl^-$ concentrations.

The formation of $NaNO_3$ through the reaction of $HNO_3$ or $NO_x$ with sea salt has been previously observed in coastal sites (Andreae et al., 1999, 2000; Bardouki et al., 2003; Dasgupta et al., 2007)(Kutsuna & Ibusuki, 1994), and within the UK and Ireland, where the interaction with marine air with polluted air masses at coastal sites was shown to significantly shift the

aerosol $NO_3^-$ to the coarse mode (Yeatman et al., 2001; Twigg et al., 2015). Scavenging of atmospheric $H_2SO_4$, formed from $SO_2$ (O'Dowd and de Leeuw., 2007), by sea salt may also be occurring, which would also shift some of the $SO_4^{2-}$ from the fine to the coarse mode.

### 4.2 Deposition velocities and fluxes of water-soluble aerosol and trace gas species

#### 4.2.1 Fluxes of water-soluble aerosols and trace gases

Fluxes of $SO_4^{2-}$ and $Cl^-$ throughout the campaign were unidirectional deposition. However, during the fertilisation period of the South Field, bidirectional fluxes of $NH_4^+$ and $NO_3^-$ were observed. Prior to fertilisation these species were deposited to the site. An apparent emission flux of $NO_3^-$ is consistent with the possibility of $NH_4NO_3$ formation above grassland suggested by the divergence of $HNO_3$ $V_d$ from $V_{max}$ (Nemitz et al., 2009b) in the presence of high concentrations of $NH_3$ near the surface. Concentrations of $NH_3$ peak at 21.4 µg m$^{-3}$ on the 13$^{th}$ June, 11:00, which occurs three hours before peak $HNO_3$ $V_d$ and 7 hours

prior to the apparent peak in emissions of $NO_3^-$ at 18:00.

Fluxes for the trace gases were bi-directional for $NH_3$ and HONO, with deposition for all other species measured. Emissions of $NH_3$ and HONO occurred throughout the campaign, with HONO emissions particularly present during the early morning. Both species reached peak emissions soon after fertilisation. Increases in atmospheric $NH_3$ concentration and emissions of

$NH_3$ resulting from the application of solid urea fertiliser has been previously established (Akiyama et al., 2004; Sommer and Hutchings, 2001), with losses from volatilisation increased if the urea pellets are poorly mixed into the soil and if conditions are dry and warm. While conditions prior to the fertilisation event were cool, temperatures increased quickly throughout the

day, peaking at 19.2 °C at 13:00, four hours after fertilisation. Volatilisation was likely exacerbated by the dry conditions throughout the 13th June. The increase in concentration and upward flux of $NH_3$ provides the source for the formation of $NH_4NO_3$ in the presence of $HNO_3$. The mechanisms of the HONO emission fluxes are not discussed here but can be found in Di Marco *et al. (*in preparation*).*

**4.2.2 Aerosol deposition velocities**

Deposition velocities for $NO_3^-$ reached a maximum value of 9.8 mm s$^{-1}$ during daytime, and a minimum of 0.2 mm s$^{-1}$ outside the period of apparent emission fluxes at night. A similar pattern was observed for sulfate, which reached a maximum value of 9.5 mm s$^{-1}$ during daytime and a minimum value, outside of apparent emission events, of 0.15 mm s$^{-1}$ during night. Median $V_d$ values for $NO_3^-$ and $SO_4^{2-}$ were 1.52 mm s$^{-1}$ and 1.45 mm s$^{-1}$, respectively. For Cl$^-$, the median $V_d$ was 3.14 mm s$^{-1}$. The deposition velocities for $SO_4^{2-}$ where larger than those previously observed and derived for accumulation mode particles from theoretical considerations (Petroff et al., 2008). For sulfate in the fine (<0.1 µm diameter) range, Allen et al. (1991) recorded a mean value of 1 mm s$^{-1}$ for deposition velocity over short grass, similar to observations made by Gallagher et al. (2002) who reported a mean value of 0.9 mm s$^{-1}$.

The dry deposition of particles can be modelled using a process-orientated approach, which describes the deposition velocity as a function of particle size based on removal mechanisms acting within the vegetation canopy, such as Brownian diffusion, impaction, interception and sedimentation (Slinn and Slinn, 1980; Davidson et al., 1982; Slinn, 1982). The models predict that for particles >0.1 µm in diameter deposition velocity increases with increasing particle size. Vong et al. (2004) recorded deposition velocities of greater than 2 mm s$^{-1}$ for $PM_{10}$ particles over grassland. If the sulfate and chloride were in particularly coarse particles, deposition velocities would potentially be skewed towards a higher deposition velocity.

Secondary ammonium compounds are typically found in the accumulation mode (0.1 to 1 µm), while seasalt is found in super-micron particles (Myhre et al., 2006). Although measurements of particle size were not made during this campaign, measurements of aerosol species (including Cl$^-$ and $SO_4^{2-}$) in the $PM_{2.5}$ and $PM_{10}$ size fractions were taken by a two-channel Monitor for Aerosols and Gases in Ambient Air (MARGA, Applikon B.V, The Netherlands) instrument located at Auchencorth Moss, 12 km south west of Easter Bush. Aerosol concentration data was taken from an online database of MARGA measurements (DEFRA, 2018). Agreement between MARGA and GRAEGOR aerosol concentrations were excellent (with correlations for $SO_4^{2-}$ with $R^2 = 0.95$, and for Cl$^-$ with $R^2 = 0.91$ between MARGA $PM_{10}$ and GRAEGOR TSP). As proxy for a particle size measurement, the proportion of $PM_{2.5}$ to $PM_{10}$ was used, with a lower proportion of $PM_{2.5}$ indicating a greater proportion of coarse aerosol, and a corresponding larger deposition velocity based on process-orientated modelling. To a first–order approximation, particle deposition velocities scale with $u_*$ (Pryor et al, 2008). Figure 12 shows plots of the normalised deposition velocities ($V_d / u_*$) against the fraction of the $PM_{10}$ mass contained in $PM_{2.5}$ at Auchencorth Moss ($f_{PM2.5} = PM_{2.5}/PM_{10}$) for nitrate (a), sulfate (b) and chloride (c).

While the dynamic range of $f_{PM2.5}$ varied between compounds, third-order polynomial curves consistently describe the relation between the proportion of $PM_{2.5}$ to overall PM and the normalised $V_d$ for nitrate, sulfate and chloride, suggesting – in line with Slinn (1982) – that deposition velocity increases strongly with increasing particle size above 0.1 µm particle diameter. How-

ever, the relationship – although statistically significant – shows significant variability, which may be due to measurement uncertainty, but might also reflect the additional effect of atmospheric stability on particle fluxes (e.g. Wesely et al., 1985; Petroff et al., 2008) or differences in the size distribution between the Auchencorth and Easter Bush measurement sites. It must be stressed that the proportion of $PM_{2.5}$ to $PM_{10}$ is a proxy measurement for particle size and can only differentiate the proportions of aerosol of diameter less than or greater than 2.5 µm.

By contrast, the median deposition velocity of 0.37 mm s$^{-1}$ for $NH_4^+$ was much smaller and within the range of previous measurements of dry deposition velocities of accumulation mode particles to grassland. The average $f_{PM2.5}$ for $NH_4^+$ recorded was 96%, compared to 78% for $NO_3^-$ and 86% for $SO_4^{2-}$, suggesting that virtually all of the $NH_4^+$ measured was contained in fine particles, within the measurement error. The average normalised deposition velocity ($V_d/u_*$) of $NH_4^+$ of 0.04 was in the

range of the values for the other compounds evaluated at $f_{PM2.5} = 100\%$.

Thus, the relatively high deposition velocities for Cl$^-$, $NO_3^-$ and $SO_4^{2-}$ (compared with $NH_4^+$) are a result of some of these compounds being contained in coarse aerosol. This is consistent with the ion balance (Fig. 11), which suggests that some of these compounds are balanced by seasalt Na$^+$, which is found mostly in the coarse fraction.

It should be noted that the increase in $V_d$ with increasing contribution of coarse aerosol only accounts for the size-dependence of the processes of impaction and interception. As a non-turbulent process, gravitational sedimentation is not reflected in micrometeorological flux measurements and the sedimentation velocity would need to be added to the deposition velocity derived here.

**4.2.3 Trace gas deposition velocities**

Median diel deposition velocities for $HNO_3$ and HCl closely matched the theoretical maximum deposition velocities within the uncertainty of the measurement (Fig. 8), which closely conforms to their expected physico-chemical behaviour. Both $HNO_3$ and HCl are reactive and highly water soluble, and consequently it is expected that their deposition velocities should equal the theoretical maximum, and that the canopy resistance for these species should be equal to zero (Dollard et al., 1987; Muller et

al., 1993). The close agreement between $V_d$ and $V_{max}$ during most of the campaign suggests that chemistry is not affecting the fluxes and that the standard aerodynamic gradient method is therefore applicable.  However, significant deviations of the calculated deposition velocity from the theoretical maximum for $HNO_3$ exist: $R_c$ values for $HNO_3$ were particularly large 40 hours after fertilisation, from the 15$^{th}$ June to the 16$^{th}$ June, when the mean $R_c$ value was 14.8 s m$^{-1}$. Conversely, there were

periods when the $V_d$ for $HNO_3$ exceeded the $V_{max}$, such as on the 13[th] June at 13:00 hours, when $V_d$ for $HNO_3$ was recorded as 56.8 mm s$^{-1}$ compared with a calculated maximum of 17.5 mm s$^{-1}$.

Reductions in $V_d$ for $HNO_3$ (or in other words a non-zero $R_c$) have been linked to ground-level sources or non-zero vapour
pressures of $HNO_3$ over nitrate-containing aerosol (particularly, $NH_4NO_3$), which may evaporate from aerosol within the air space below the measurements or previously deposited to leaf surfaces (Brost et al., 1988; Kramm and Dlugi, 1994; Nemitz et al., 2000a, 2004). By contrast, values of $V_d$ for $HNO_3$ that exceed the theoretical maximum could suggest the presence of an additional sink for $HNO_3$, which would potentially arise as the result of $NH_3$ reactions with $HNO_3$ to form $NH_4NO_3$ (Nemitz et al., 2000b; van Oss et al., 1998). The higher $V_d$ values for $HNO_3$ during the fertilisation period, followed by a higher $R_c$
value 40 hours afterwards, could suggest the formation of $NH_4NO_3$ immediately following fertilisation followed by its volatilisation soon after. Indeed, the exceedance of $V_{max}$ coincided with upward fluxes of $NH_4^+$ and $NO_3^-$ (Fig. 9) and this suggests that during the period after fertilisation, the increase in $NH_3$ concentration lead to an exceedance of the equilibrium vapour pressures of $NH_4NO_3$ near the ground, resulting in partitioning of $NH_3$ and $HNO_3$ into the aerosol phase. This would have constituted an additional airborne sink for $HNO_3$ ($V_d > V_{max}$) as well as a source (apparent emission) for $NH_4^+$ and $NO_3^-$ as
previously reported by Nemitz et al. (2009).

It should be noted that during this period the aerodynamic gradient method does not derive accurate fluxes because the condition of flux conservation is not met (Wolff et al., 2010), and this period has therefore not been included in the diel cycles and summary statistics presented above.

By contrast, fluxes of total ammonium (tot-$NH_4^+$ =$NH_4^+$ + $NH_3$) and total nitrate (tot-$NO_3^-$ = $NO_3^-$ + $HNO_3$) would be conserved, as the effect of gas-particle interactions are not considered, and their assessment provides additional information on the processes occurring during periods when fluxes are not conserved with height.

The time series for tot-$NO_3^-$ and tot-$NH_4^+$ fluxes are shown in Figure 13. Prior to the fertilisation event on the 13[th] June, the fluxes for tot-$NO_3^-$ were universally depositional to the surface, while fluxes of tot-$NH_4^+$ were bi-directional with significant variation. However, six hours after fertilisation, a significant emission event of tot-$NO_3^-$ was observed lasting for six hours. Interestingly, this indicates that the apparent $NO_3^-$ emission during this period (Fig. 9) exceeded the measured deposition of $HNO_3$, and that there must have been a net source of $NO_3^-$ at the surface during this period. Upward fluxes have previously
been reported in the literature where it was attributed to the volatilisation of $NH_4NO_3$ from leaf surfaces (Neftel et al., 1996) or alkyl nitrate chemistry (Farmer and Cohen, 2008). Primary emissions of $HNO_3$ could arise from the heterogeneous reaction of $NO_2$ with water (Harrison, 1996):

$$2NO_2 + H_2O \longrightarrow HONO + HNO_3, \hspace{4cm} (R3)$$

Kleffman (2007) suggests that $HNO_3$ could be formed by the reduction of $NO_2$ on organic sources of humic acid, a process that would also lead to the production of HONO. The formation of $HNO_3$ inferred from observations coincided with emissions of HONO post–fertilisation. However, as discussed previously, this reaction is slow, and while possibly contributing to some of the observed HONO emission, may not be able to account for the majority of observed emissions.

A second potential pathway is the emission of HONO from the soil. As described by Scharko et al. (2015), the oxidation of ammonium by microbes in soils with high nitrification rates can lead to biogenic emissions of HONO. The addition of urea to the agricultural soil at Easter Bush would lead to an increase in soil $NH_4^+$ concentrations and subsequently, through oxidation by soil microbes, the observed emission of HONO. Further discussion of the sources of HONO emissions at Easter Bush will

be described in a future paper by Di Marco *et al.* (in preparation).

## 4.3 Daytime Source of HONO

As shown in Figure 5, the median diel concentrations for HONO recorded by the GRAEGOR at 2.4 m do not drop below the detection limits of the instrument, determined to be 30 ng m$^{-3}$ from calibrations carried out during the campaign. This is contrary to what would be expected based solely on the photolysis rate of HONO, which would suggest that, after accumulation

of HONO during night–time, rapid photolysis should reduce concentrations to below the detectable levels for measurement during early morning (Pagsberg et al., 1997). As measurement approaches have improved over the past 10 years, a growing number of measurements have revealed non-negligible HONO daytime concentrations at rural (Acker et al., 2006; Su et al., 2008; Sörgel et al., 2011), agricultural (Laufs et al., 2017) and urban (Lee et al., 2016) sites, including previous studies at the Easter Bush site (Twigg et al., 2011). Details on the discussion of a potential daytime source of HONO are further discussed

in Di Marco *et al.* (in preperation).

## 4.4 Comparison of GRAGOR with other instrumentation

### 4.4.1 Comparison of nitrous acid measurement between GRAEGOR and LOPAP

The comparison between the LOPAPs and the GRAEGOR revealed that both sample boxes of the GRAEGOR measured higher HONO concentrations than the LOPAP, principally due to the presence of a constant concentration offset of 0.01 to

0.02 µg m$^{-3}$ of HONO. Previous comparisons of measurements of HONO have been between the wet annular rotating denuder (WRD), as used in the GRAEGOR, and optical absorption techniques, primarily differential optical absorption spectroscopy (DOAS) instruments. In those comparisons, it has been found that HONO measurements by WRD, particularly during daytime and at low concentrations, tend to be significantly higher than DOAS measurements (Appel et al., 1990). By comparison, the LOPAP shows good agreement in HONO measurements with the DOAS (Kleffman et al., 2006), as the DOAS method is a

molecule specific method and the LOPAP method measures any potential NOx artefact.

The higher concentrations recorded by the GRAEGOR can be explained by the presence of chemical interferences that occur on the inlet, at the air/liquid interface and within the sampling solution. As the WRD uses a liquid film to sample HONO, and as HONO can form heterogeneously on such surfaces, overestimation of HONO can occur. Furthermore, interferences by chemical reactions of $NO_2$ with hydrocarbons within the sampler can lead to a further interference (Gutzwiller et al., 2002),

particularly in proximity to diesel emissions. It has also been shown that in high-alkalinity sampling solutions, mixtures of $SO_2$ and $NO_2$ can add a further interference to measurements (Spindler et al., 2003). Finally, photolytically induced artefacts can be introduced in the sampling lines that connect the GRAEGOR sampling box to the detector unit (Kleffman and Wiesen, 2008). The LOPAP, which is also a wet chemistry-based instrument, is designed to minimise the chemical interferences and artefacts that can be introduced in other wet chemistry instruments.

A comparison between daytime (06:00 to 18:00) GRAEGOR HONO concentrations and LOPAP HONO concentrations found only a slightly greater difference than the comparison between night time (19:00 to 05:00) concentrations recorded by the GRAEGOR and LOPAP. While previous comparisons between the DOAS and the WRD found that daytime concentrations measured by the WRD were higher than the DOAS compared to night-time measurements, these studies were generally con-

ducted in urban areas where both HONO and $NO_x$ concentrations were high (Febo et al, 1996), in contrast to the low concentrations at Easter Bush. The implementation of thermal insulation material around the GRAEGOR sampling lines may have also reduced the influence of photolytic artefacts in exposed sampling lines during the day, which would have elevated daytime HONO measurements recorded by the GRAEGOR.

Spindler et al. (2003) developed the following quantification of the chemical artefact produced by the mixing of $NO_2$ and $SO_2$ in highly alkaline sampling solutions for HONO measurements in their investigation of $SO_2$ and $NO_2$ chemical interference, with all concentrations measured in ppb.

$$[HONO]_{artefact} = 0.0056[NO_2] + 0.0022 \text{ ppb}^{-1}[NO_2][SO_2] \qquad (12)$$

The first term describes the heterogeneous formation of $NO_2$ with water alone, and the second describes the aqueous-phase

reaction of $NO_2$ and $SO_2$. Using measurements of $SO_2$ and $NO_2$ concentrations, the HONO artefact for the period of the GRAEGOR-LOPAP comparison was calculated and subtracted from the HONO concentrations recorded by the GRAEGOR. A linear regression between the concentrations recorded by GRAEGOR Sample Box 2 and LOPAP (2.0 m), which had the best agreement without artefact reduction, indicated better agreement after the correction for the artefact (GRAEGOR$_{artefact}$(2.4 m) = 1.02*HONO(HONO(LOPAP(2.0 m)), intercept = 5 x $10^{-3}$ µg m$^{-3}$, $R^2$ = 0.72). Coefficient values were altered to produce

the best possible agreement between GRAEGOR and LOPAP HONO values, arriving at a final artefact quantification of:

$$[HONO]_{artefact} = 0.0090[NO_2] + 0.0034 \text{ ppb}^{-1}[NO_2][SO_2] \qquad (13)$$

Use of these altered coefficients further reduced the offset in GRAEGOR HONO measurements, but also reduced the statistical agreement between GRAEGOR and LOPAP HONO measurements (GRAEGOR$_{artefact}$(2.4 m) = 0.98*HONO(HONO(LOPAP(2.0 m)), intercept = 2 x $10^{-3}$ µg m$^{-3}$, $R^2$ = 0.57). Figure 14 shows the results of these analyses, with the linear regression between GRAEGOR Sample Box 2 (2.4 m) and LOPAP (2.0 m) without the artefact reduction applied to GRAEGOR Sample Box 2 (2.4 m) HONO concentrations, with Spindler's artefact reduction, and with the modified Spindler's artefact reduction. While the agreement between the LOPAP and GRAEGOR is improved by the introduction of an artefact reduction value, it does not fully close the gap even with altered coefficient values, with a constant concentration offset still present in measurements. The possibility that a further artefact is introduced from $NO_2$ mixing with hydrocarbons would require further investigation, with concurrent measurements of hydrocarbons.

To determine if the HONO concentration offset in the GRAEGOR measurements impacted upon the measurements of HONO flux, a comparison between the HONO flux values derived from GRAEGOR and LOPAP measurements was conducted. Concurrent fluxes of HONO derived from LOPAP and GRAEGOR measurements exist for 72 hourly measurements, from the 26$^{th}$ May – 6$^{th}$ June. Figure 15 shows (a) the full time series of concurrent HONO flux values derived from GRAEGOR and LOPAP measurements and (b) a scatter plot of GRAEGOR against LOPAP HONO flux values. Overall, GRAEGOR HONO fluxes are biased towards deposition, with greater deposition values and lesser emission values compared to concurrent LOPAP values. This pattern would be consistent with the concept of an artefact formation dependent upon $SO_2$ and $NO_2$. $SO_2$ fluxes were unidirectionally depositional at Easter Bush during the campaign as measured by the GRAEGOR. Deposition of $SO_2$ would lead to greater formation of artefact within the sample box set at a higher height, which is consistent with comparisons of HONO concentrations between each sample box of the GRAEGOR and each LOPAP instrument. In turn, this would lead to a bias in HONO flux values, resulting in a skew towards deposition. It should be noted that the sample size of concurrent measurements of HONO flux from GRAEGOR and LOPAP measurements is limited (n = 72).

### 4.4.2 Comparison of ammonia measurements with GRAEGOR and QCL

The comparison between the GRAEGOR and QCL found that, while there was reasonable agreement between the instruments, the GRAEGOR measured somewhat higher $NH_3$ concentrations than the QCL, by a factor of 1.2. Due to lack of ancillary micrometeorological data during this campaign, the short overlap in measurements, and necessary filtering of unreliable data, there are too few concurrent measurements (15 hours) of flux between the QCL and the GRAEGOR for a reliable comparison. There are also only 41 hours of concurrent concentration measurements between the two instruments, which overlapped with a period of low $NH_3$ concentrations.

A similar comparison between a WRD system (the Ammonia Measurement by Annular Denuder with Online Analysis, AMANDA) and the QCL system was conducted at the same site in 2004 and 2005 by Whitehead et al. (2008). This comparison also found that the WRD system measured higher concentrations of $NH_3$ compared to the QCL, but at a far greater factor of

1.67. This difference was particularly pronounced during periods of low $NH_3$ concentrations, with better agreement recorded during a fertilisation and cutting event that occurred during that study. The older (pumped) QCL used during the earlier campaign did not derive its concentrations from first principles, in contrast to the QCL used during the comparison with the GRAEGOR reported here, which should be within 3% of the absolute value without further calibration, according to the manufacturer. An inter-comparison between eleven different measurement techniques for $NH_3$ – including the AMANDA and two QCL instruments (the DUAL-QCLAS and the compact-QCLAS) - was conducted at the Easter Bush site in 2008 (von Bobrutzki et al., 2011). While good statistical agreement was found in linear regression between the AMANDA and both QCL instruments for $NH_3$ concentrations throughout the entirety of the campaign ($R^2$ =0.92 and $R^2$ = 0.97 for the compact-QCL and DUAL-QCLAS, respectively), there was less agreement between the instruments during periods of low (<10 ppb) $NH_3$ concentrations ($R^2$ = 0.81 and $R^2$ = 0.52 for the compact-QCL and DUAL-QCLAS respectively). During periods of low concentration, the QCL systems also underestimated $NH_3$ concentrations compared to the AMANDA.

Any errors in the GRAEGOR's internal $NH_3$ calibration system are unlikely to have an effect at low $NH_3$ concentrations. As a test, the calibration values obtained from all the internal calibration checks which were carried out through the campaign (total calibrations = 5) were used to calculate the $NH_3$ concentrations during the period of QCL measurements. No significant concentration difference was found between the concentrations obtained by different calibration values, due to no systematic difference in agreement between the different calibration periods.

While there remain significant differences in measured $NH_3$ concentrations between the GRAEGOR and QCL, the improved agreement between those concentrations, particularly at low values, compared with the results from 2004 and 2005 suggests an improved methodology in use by the QCL system in place at Easter Bush. Further measurements, particularly of fluxes and during periods of high $NH_3$ concentrations, would be required for a more detailed analysis.

## 5 Conclusion

In this paper, we have presented for the first time simultaneous measurements of the trace gases HCl, HONO, $HNO_3$, $SO_2$ and $NH_3$, and their associated water-soluble aerosols counterparts $Cl^-$, $NO_2^-$, $NO_3^-$, $SO_4^{2-}$, $NH_4^+$, before and after urea fertilisation of an agricultural grassland. The main findings for this study are:

1. Simultaneous measurements of the components of the $NH_3$-$NO_3$-$NH_4NO_3$ triad suggested formation of ammonium nitrate post fertilisation. The use of the conservative exchange fluxes tot-$NH_4^+$ and tot-$NO_3^-$ indicates the presence of a ground source of $HNO_3$ post fertilisation, which would be rapidly scavenged by high post-fertilisation concentrations of $NH_3$ to form $NH_4NO_3$. Through this mechanism, use of urea fertiliser becomes a source of regional, rather than local, pollution.

2. The deposition velocities measured for the aerosol compounds $Cl^-$, $NO_3^-$ and $SO_4^{2-}$ were significantly larger than those measured for $NH_4^+$. After normalisation by turbulence, the measurements suggested a clear relationship between deposition velocity and particle size for $Cl^-$, $NO_3^-$ and $SO_4^{2-}$, as parameterised using the proxy of compound in $PM_{2.5}/PM_{10}$, although the relationship shows significant variability. Therefore, the high deposition velocities for aerosol compounds recorded at the site are a result of a fraction of the compounds being contained in super-micron aerosol, such as sea–salt sulphate and sodium nitrate.

3. Evidence for a HONO daytime source at the site throughout the campaign adds to the growing body of past measurements that has found evidence for HONO daytime formation in rural, urban and agricultural areas. There is also evidence for the emission of HONO post fertilisation at the site.

This also appears to be the first time a comparison between measurements of HONO concentrations determined by the LOPAP and the GRAEGOR instruments has been documented. While good linear agreement exists between HONO measurements taken by GRAEGOR and LOPAP at both measurement heights, a consistent offset in GRAEGOR HONO measurements suggest the presence of chemically induced artefacts within the GRAEGOR system. This is potentially linked to atmospheric $SO_x$ and $NO_x$ concentrations.

Furthermore, this paper presents a comparison between measurements of $NH_3$ concentration determined by the GRAEGOR and a QCL system. While changes to the QCL operation system compared to previous studies conducted at the site have resulted in better agreement between the GRAEGOR and QCL, particularly for low $NH_3$ concentrations, there still remain significant differences in $NH_3$ concentrations with larger values reported by the denuder system.

Future measurements of aerosol deposition velocities should aim to investigate the effect of particle size upon deposition velocity, using a more robust measurement of particle size than used here. In addition, the ability of urea pellets to act as a potential surface on which heterogeneous formation of HONO and $HNO_3$ occurs should be investigated, particularly as the formation of these compounds can give rise to the formation of the regional pollutant $NH_4NO_3$.

**Acknowledgments**

This work was supported through studentship funded jointly by the Max Planck Institute for Chemistry and the University of Edinburgh School of Chemistry, and by the UK Natural Environment Research Council (NERC) through the project "Sources of Nitrous Acid in the Atmospheric Boundary Layer" (SNAABL, NE/M013405/1), with additional field-site support from NERC National Capability funding. The QCL was operated within the framework of the "UK-China Virtual Joint Centre for

Improved Nitrogen Agronomy" funded through the Newton programme and administered by the UK Biotechnology and Biological Sciences Research Council (BBSRC). Data from the MARGA was obtained from https://uk-air.defra.gov.uk/data/data_selector and is subject to Crown copyright, Defra, licenced under the Open Government Licence (OGL).

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

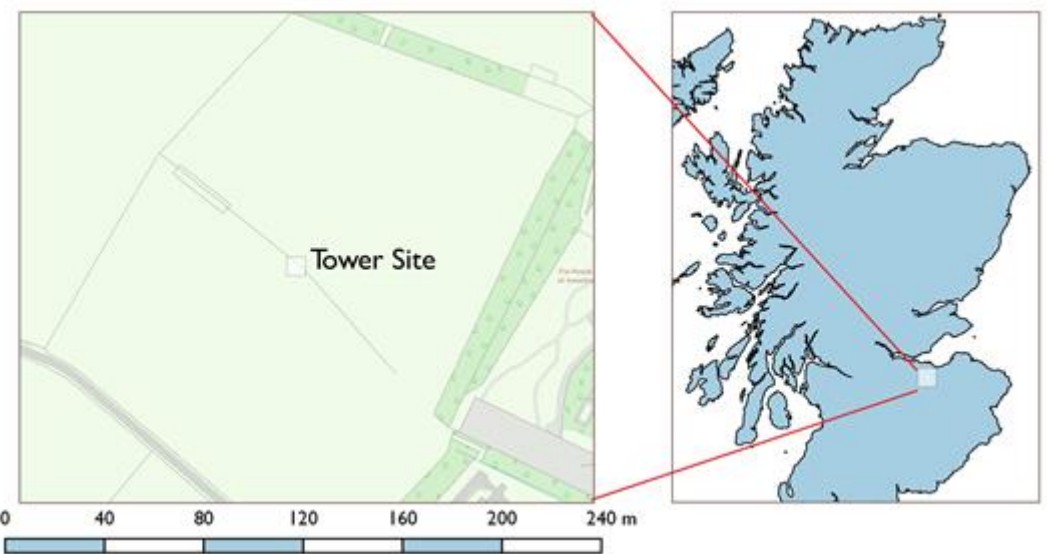

**Figure 1: Location of the Bush Tower Site (3°12'W, 55°52'N) in relation to surrounding agricultural land and within Scotland, UK.**

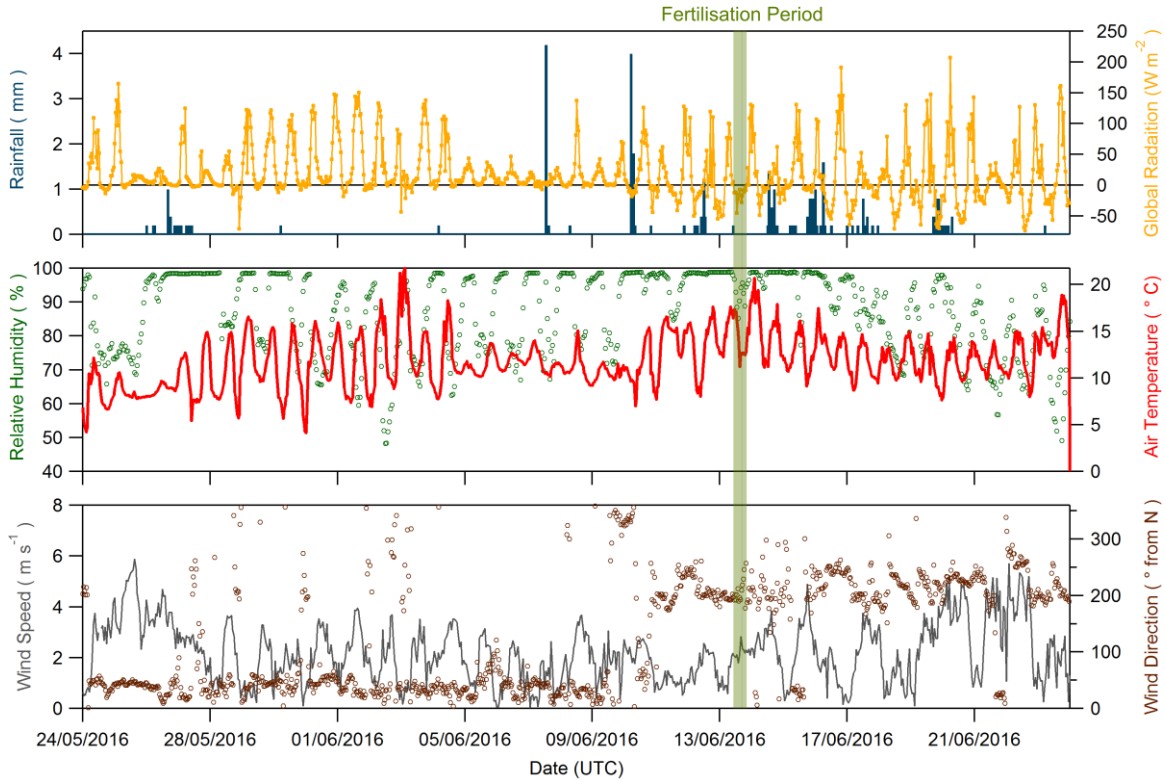

**Figure 2: Global radiation (orange line), rainfall (blue bars), relative humidity (green dots), air temperature (red line), wind direction (brown circles) and wind speed (grey line) recorded during the Easter Bush Campaign, May to June 2016. The fertilisation period was 08:00 – 09:00 on 13th June and is highlighted in green.**

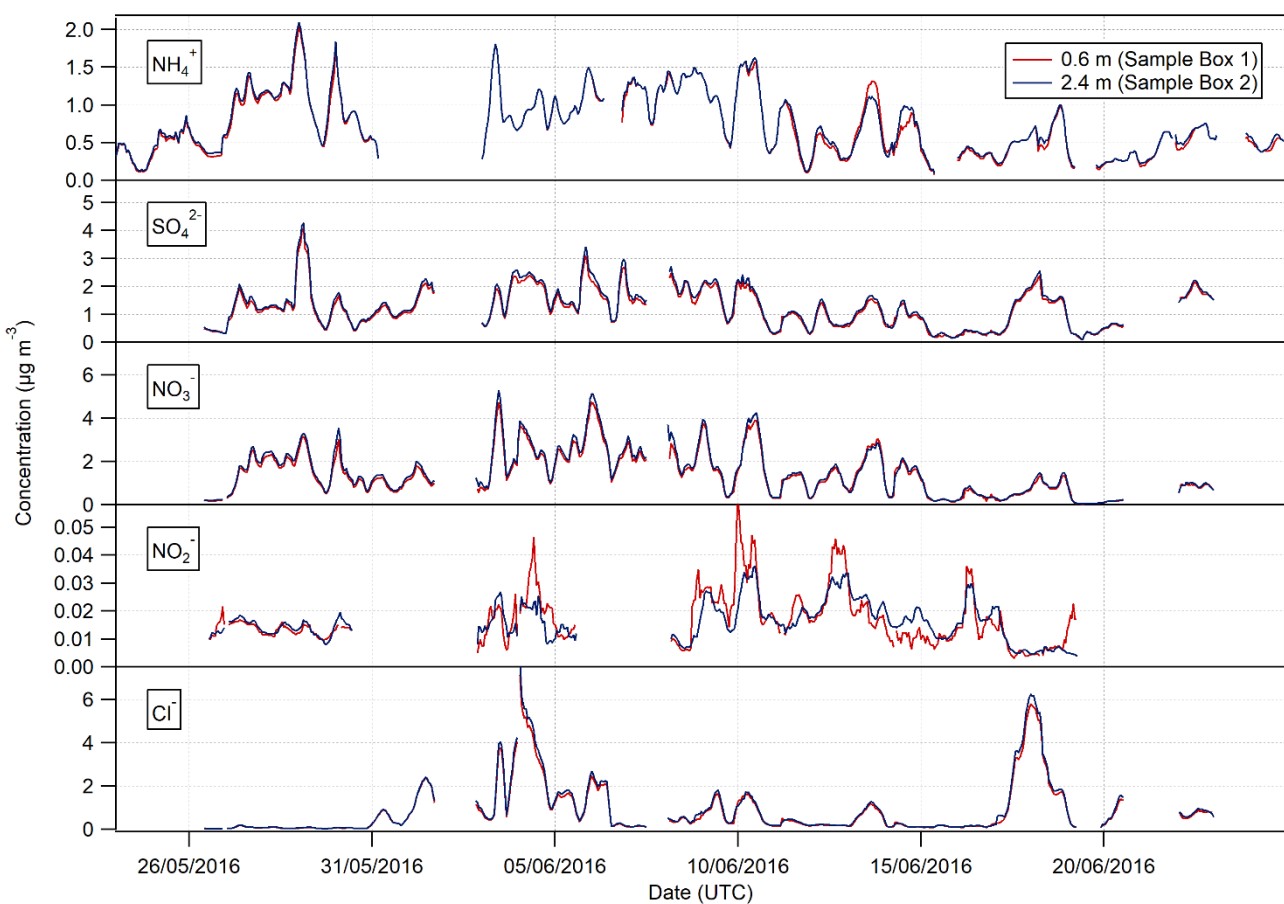

**Figure 3: Time series of hourly concentrations of the water-soluble aerosol species measured during the Easter Bush campaign. Results smoothed using a 5-hour moving point average.**

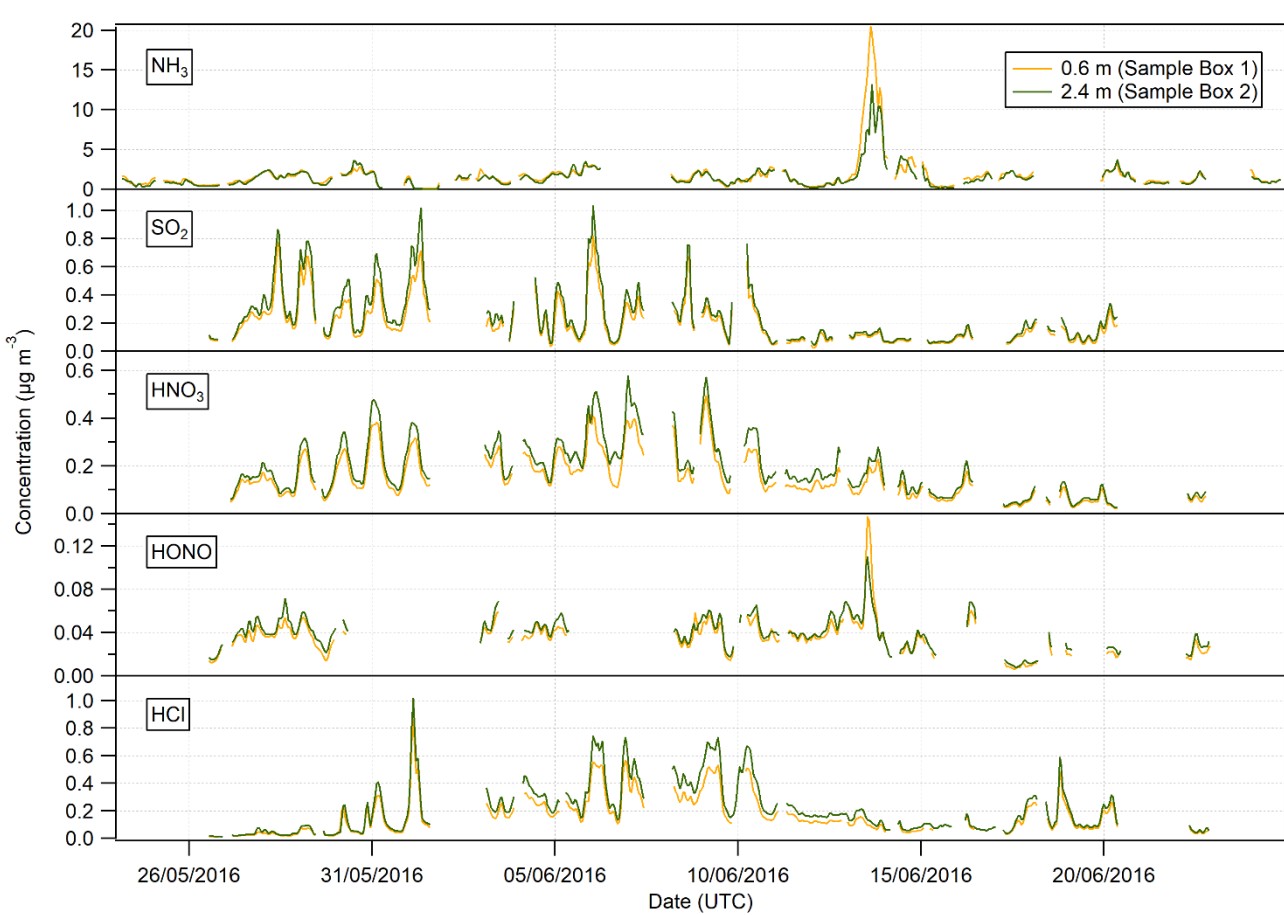

**Figure 4: Time series of hourly concentrations of the gaseous species measured during the Easter Bush campaign. Results smoothed using a 5-hour moving point average.**

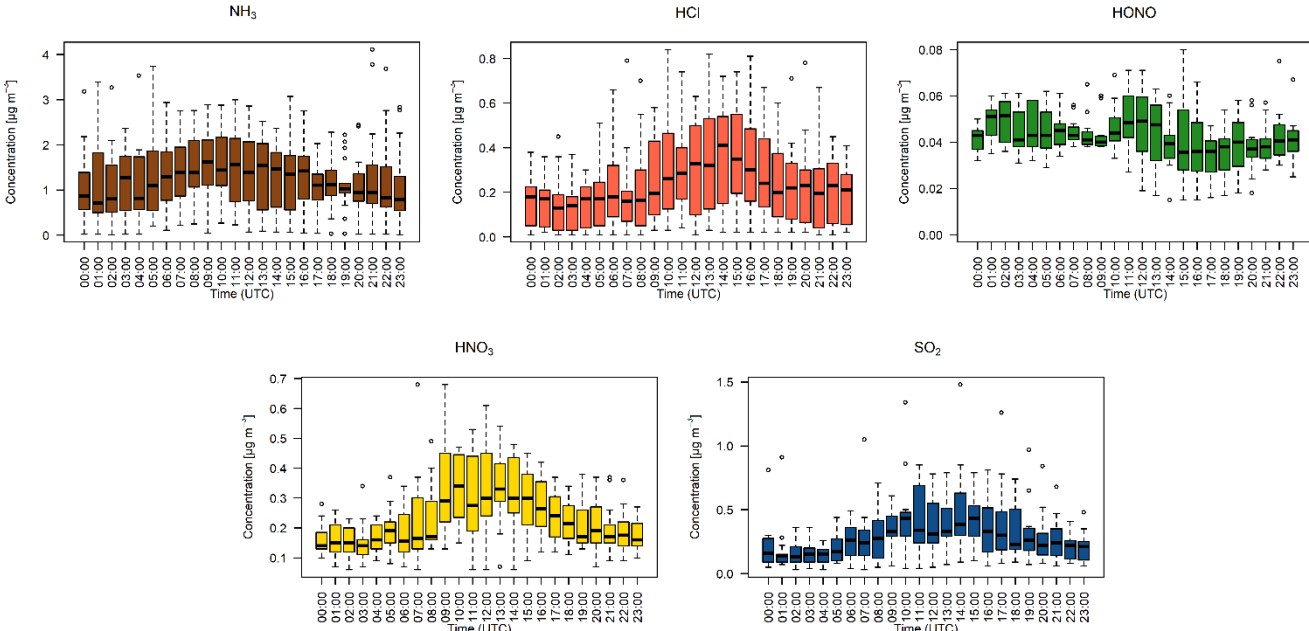

**Figure 5: Hourly median diel trace gas concentrations measured by the GRAEGOR at 2.4 m. Boxes show the lower and upper quartiles and whiskers the 5% to 95% range, with outliers shown as circles.**

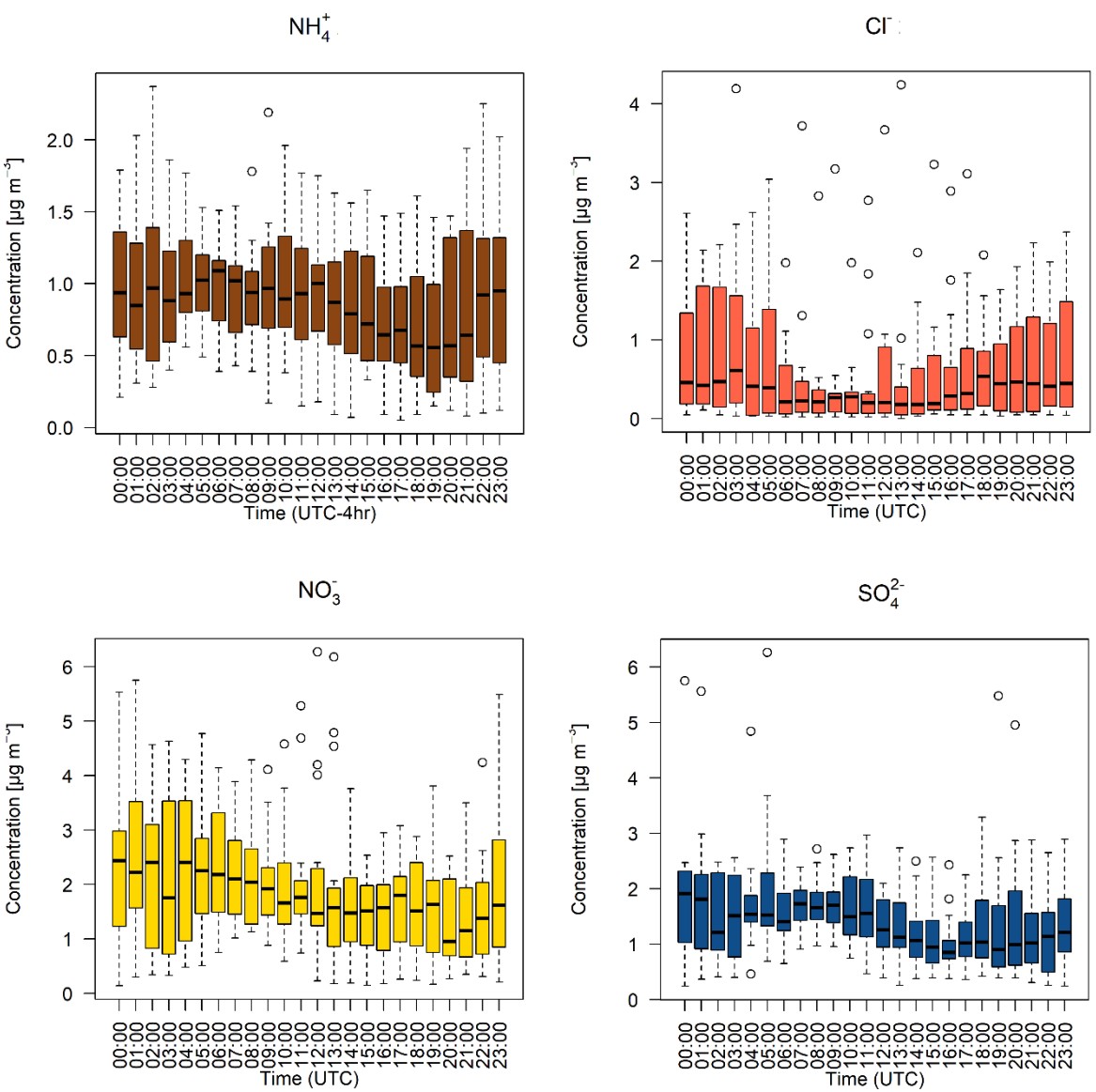

**Figure 6: Hourly median diel water–soluble aerosol concentrations measured by the GRAEGOR at 2.4 m. Boxes show the lower and upper quartiles and whiskers the 5% to 95% range, with outliers shown as circles.**

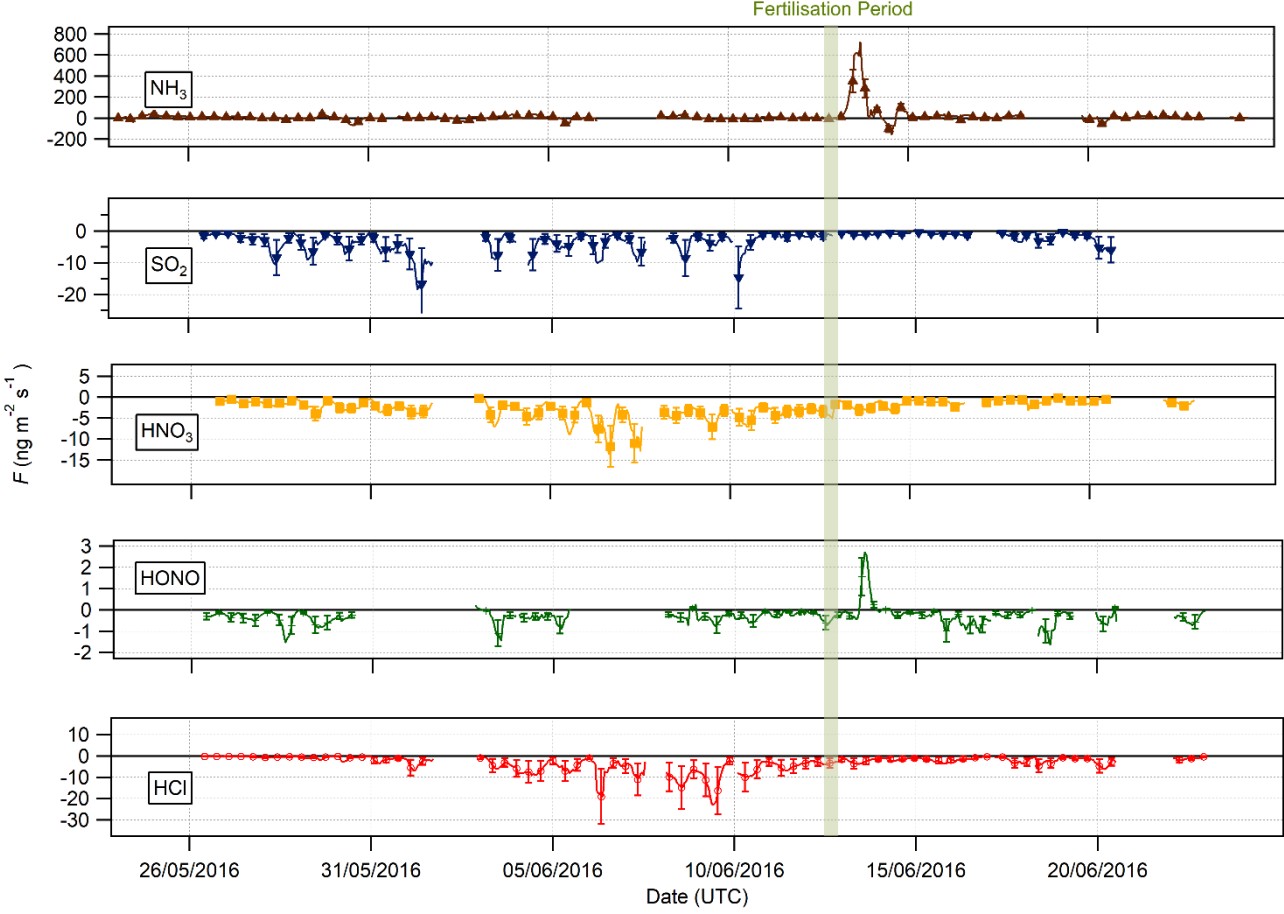

**Figure 7: Time series of hourly trace gas fluxes measured during the Easter Bush campaign. Results smoothed using a 5-point moving point average. The fertilisation period was 08:00 – 09:00 on 13th June, and is highlighted in green. Flux uncertainties for each trace gas are included as error bars.**

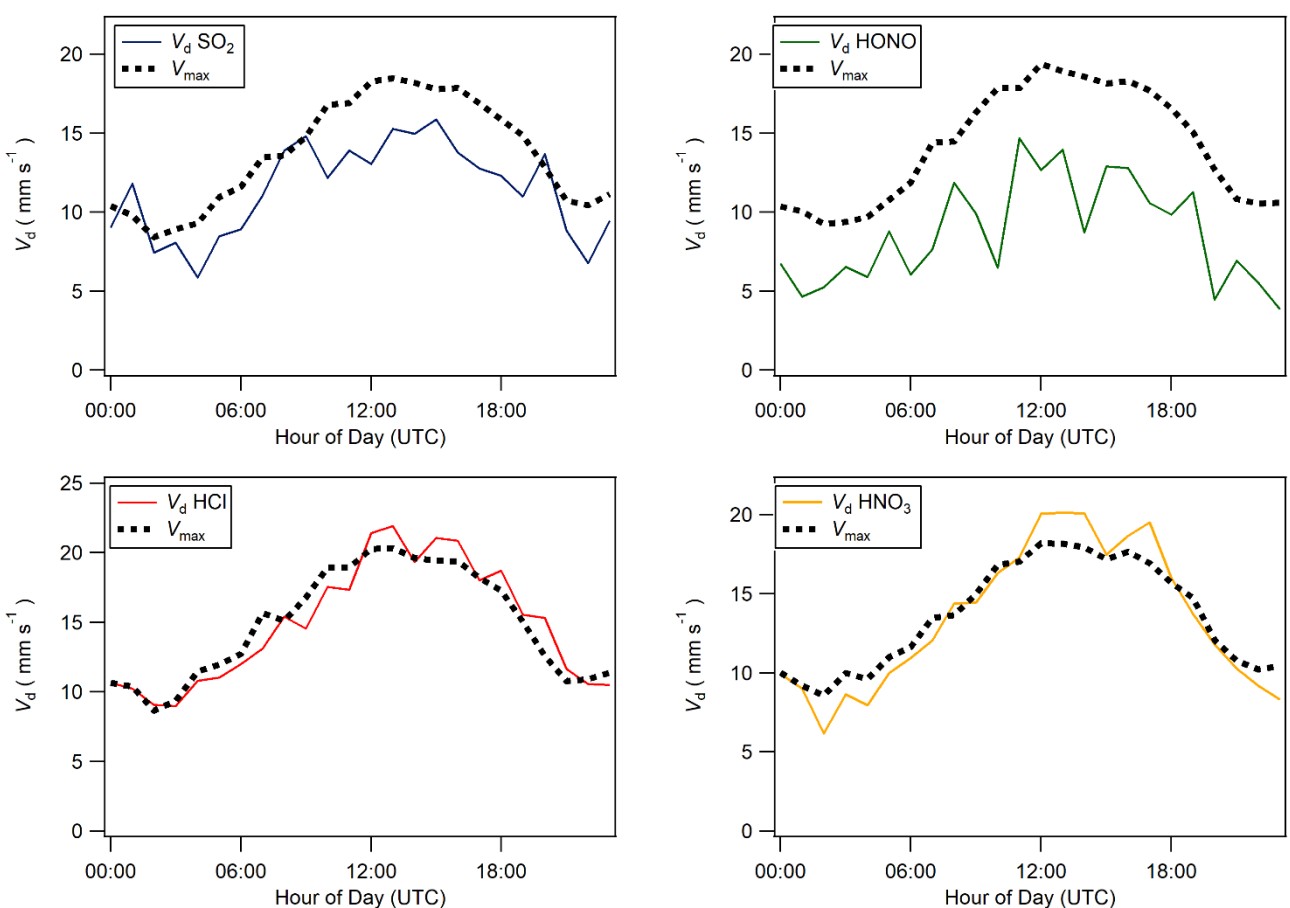

**Figure 8: Median diel cycles for deposition velocity ($V_d$) and maximum deposition velocity ($V_{max}$) for (from top left clockwise) $SO_2$, HONO, $HNO_3$ and HCl.**

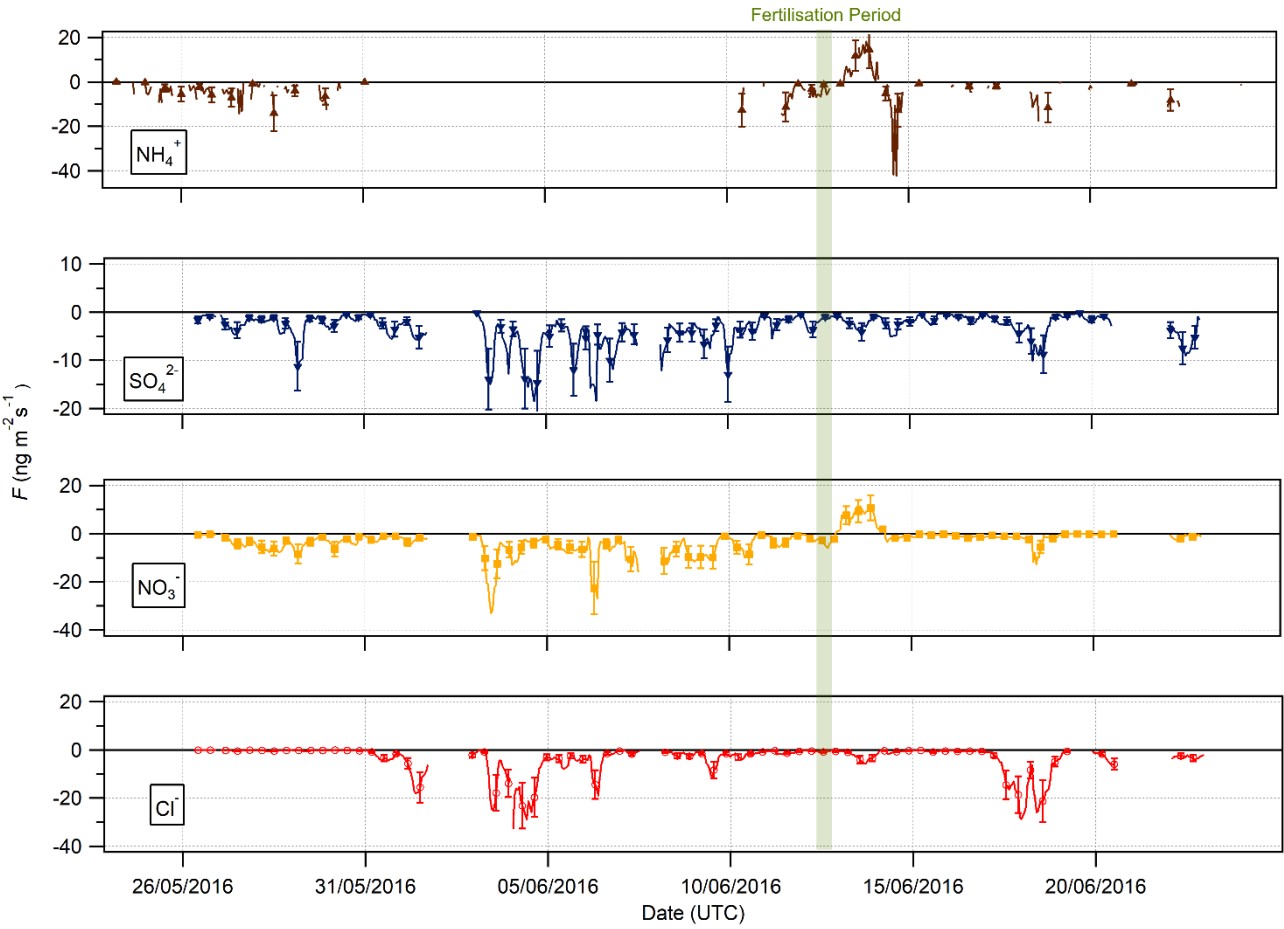

**Figure 9: Time series of hourly fluxes of water-soluble aerosol species measured during the Easter Bush campaign. Results smoothed using a 5-point moving point average. The fertilisation period was 10.00 on 13th June, and is highlighted in green. Flux uncertainties for each aerosol are included as error bars.**

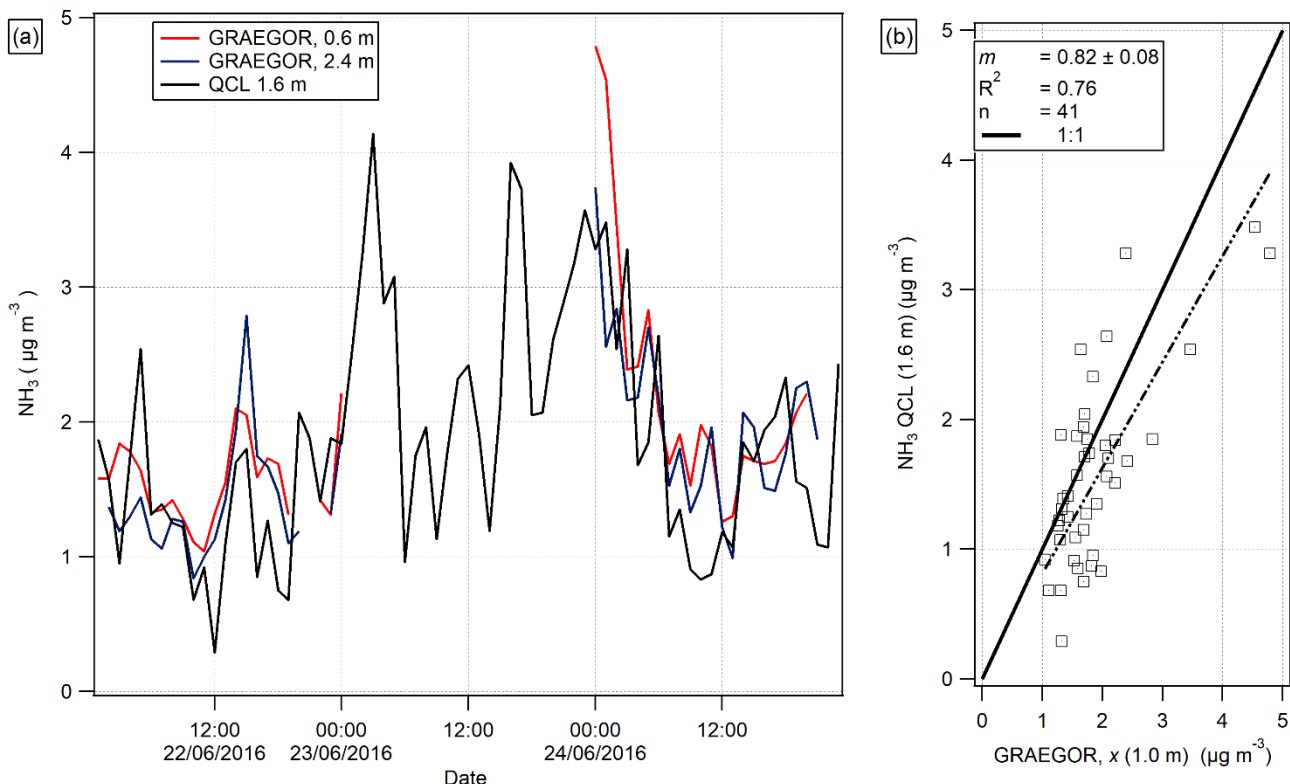

**Figure 10 : (a). Time series of hourly averages of NH₃ measurements recorded by GRAEGOR (0.6 m and 2.4 m) and QCL. (b). Linear regression analysis between QCL NH₃ measurements and GRAEGOR (derived averaged concentration at 1.0 m) NH₃ measurements.**

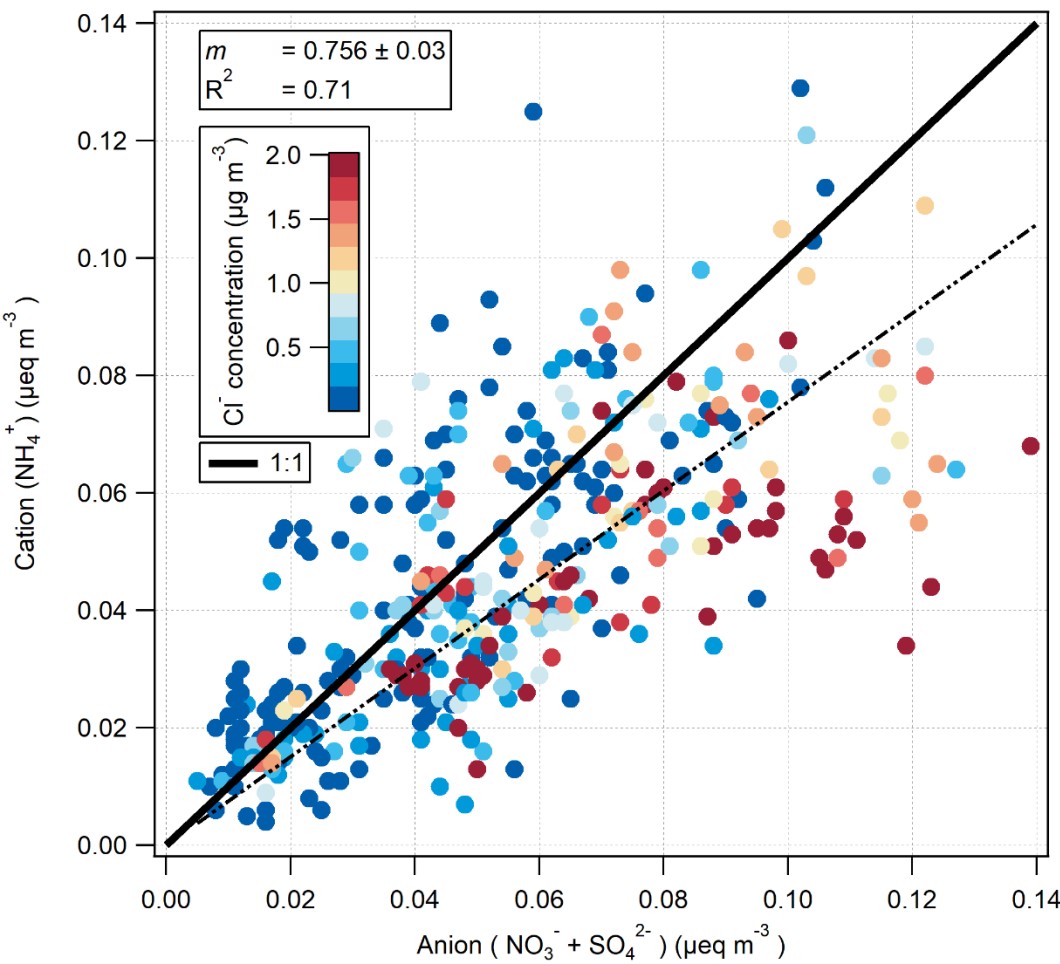

**Figure 11: The ion balance of measured selected anions ($NO_3^-$ + $SO_4^{2-}$) and measured cations ($NH_4^+$) in µeq m⁻³. The colour scale is capped at 2 µeq m⁻³ Cl⁻ to highlight the association of anion excess with periods of sea salt influence.**

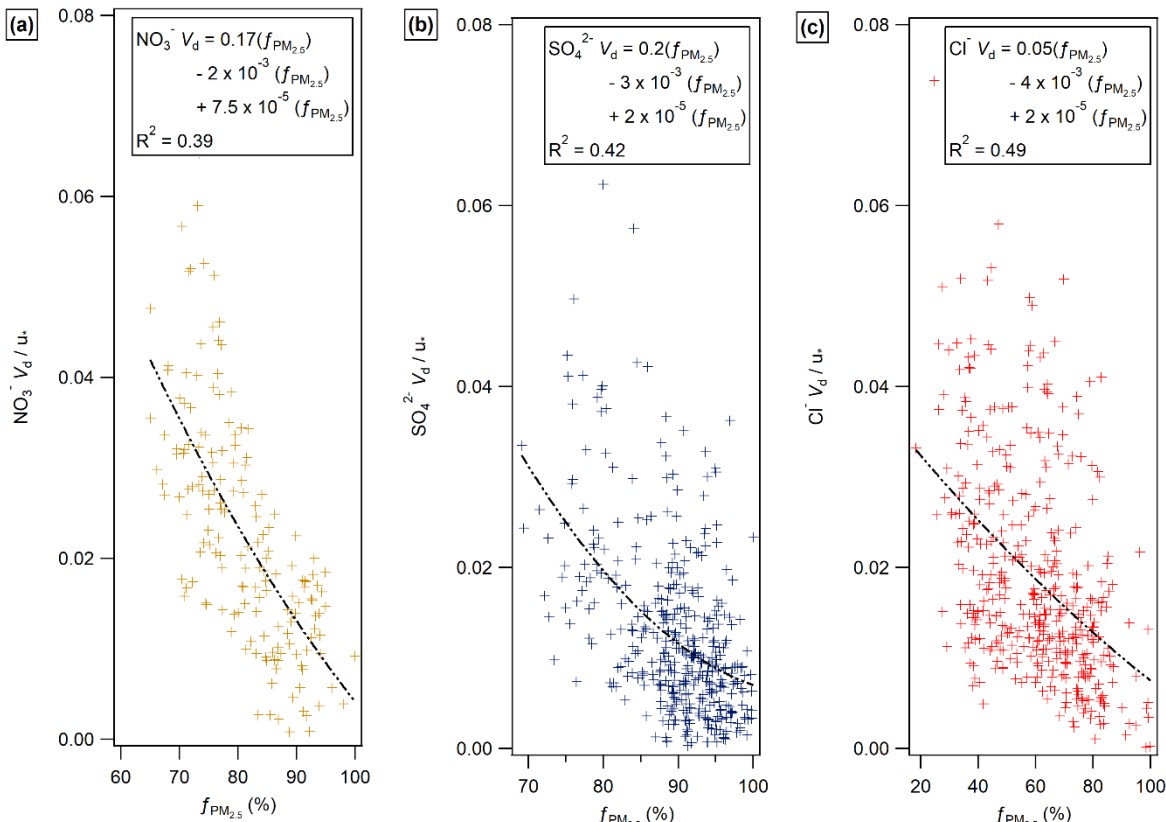

**Figure 12: The normalised deposition velocity as a function of $f_{PM2.5}$ (expressed as a %) for (a) nitrate, (b) sulfate and (c) chloride, derived from the MARGA measurements at Auchencorth Moss.**

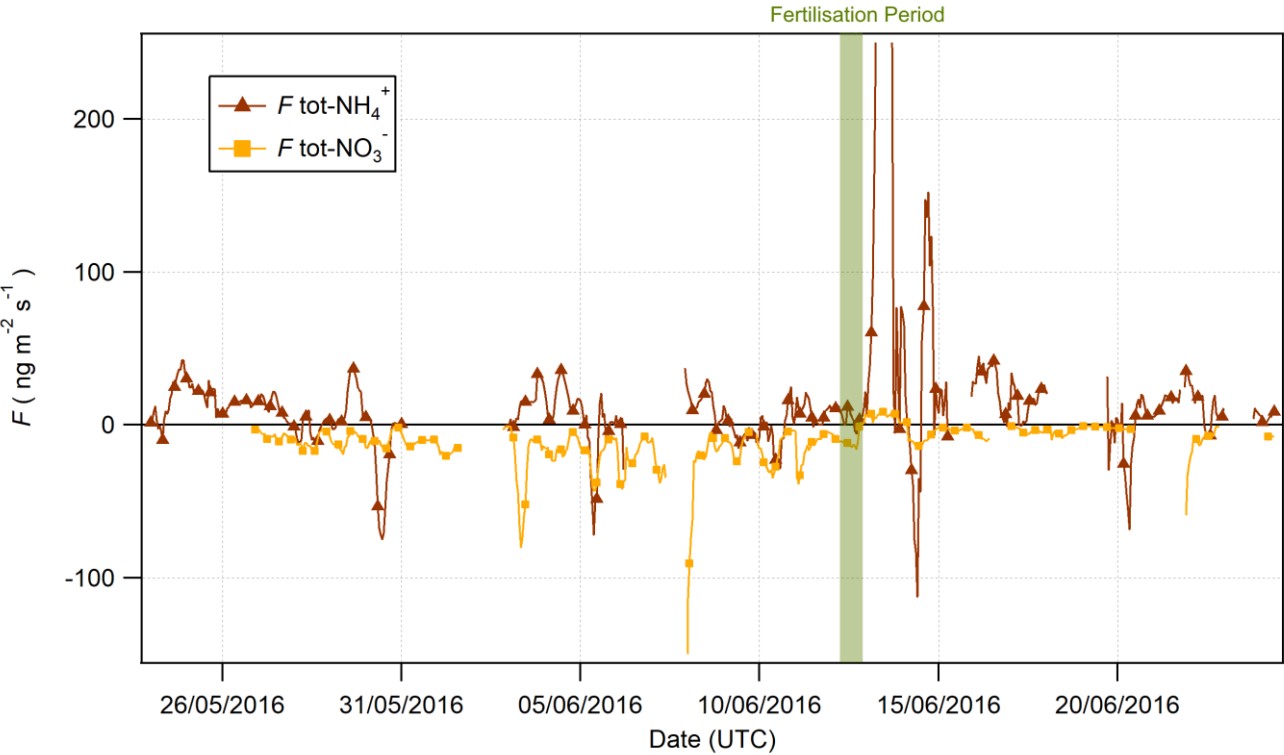

**Figure 13: Fluxes of tot-NO₃⁻ and tot-NH₄⁺ pre-and post-fertilisation on the 13th June 2016 at 09:00 (marked in green).**

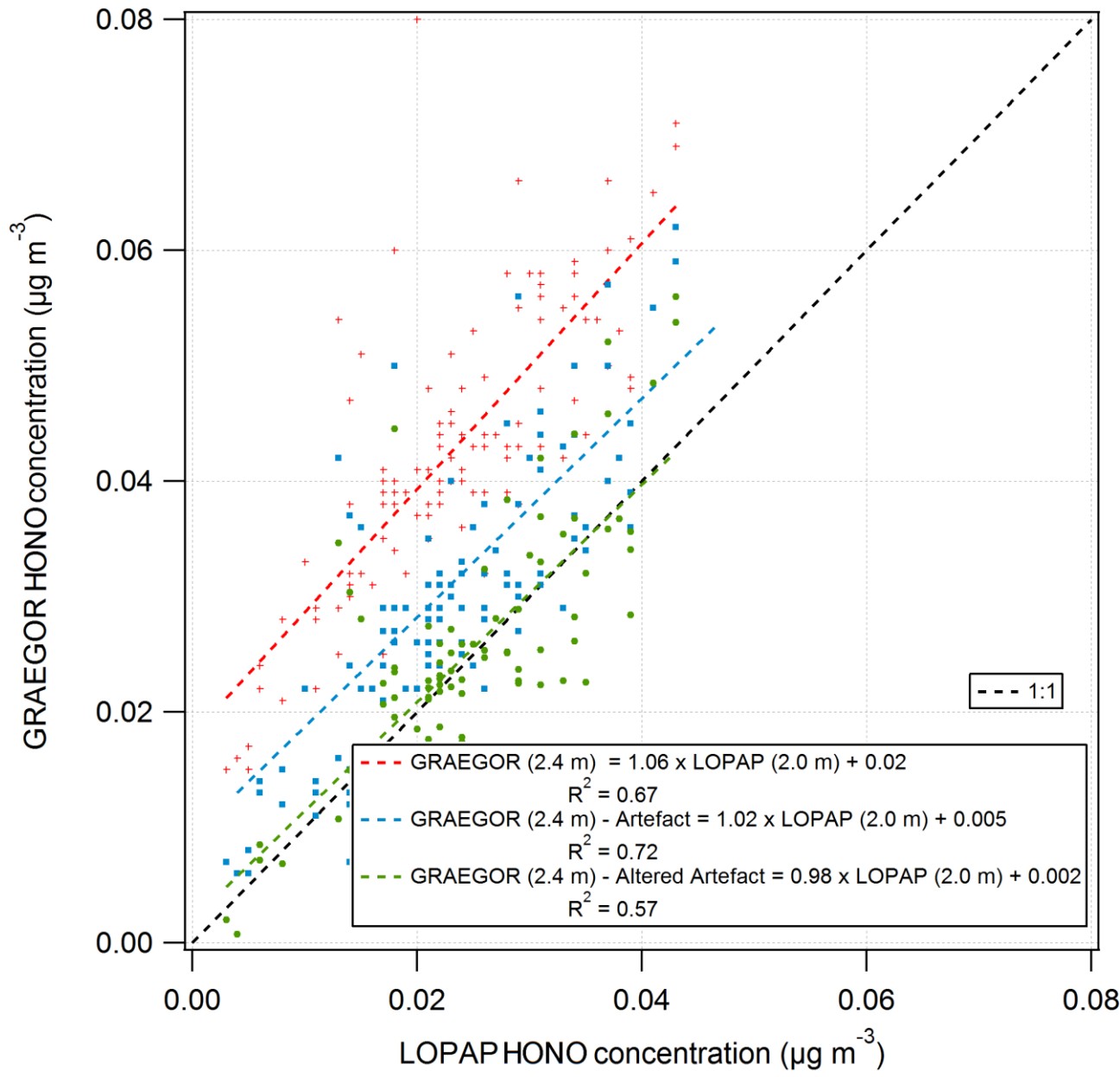

**Figure 14: Simple linear regression analyses between GRAEGOR (2.4 m) and LOPAP (2.0 m) without artefact reduction (red), with Spindler's artefact reduction (blue), and with modified Spindler's artefact reduction (green) applied to GRAEGOR (2.4 m) HONO concentration.**

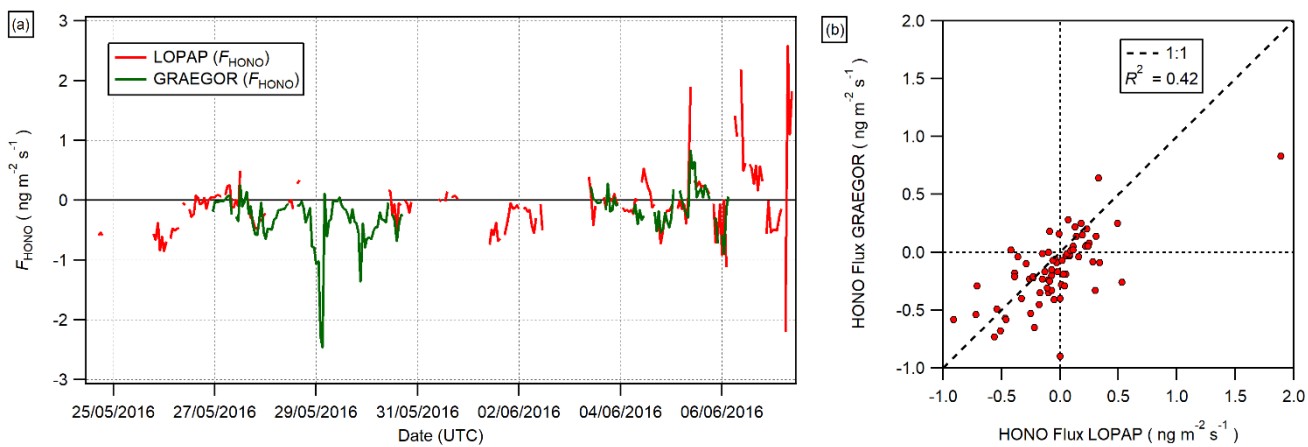

**Figure 15: (a)** Time series of concurrent flux measurements of HONO derived from LOPAP (red) and GRAEGOR (green) measurements. **(b)** Scatter plot of GRAEGOR HONO flux values against LOPAP HONO flux values.

**Table 1:** Limits of detection (LOD, determined as three standard deviations from average baseline signal), mean ($\mu_A$), median ($\mu_M$), min, max, and arithmetic standard deviation ($\sigma_A$) for concentrations measured at 2.4 m for trace gas and water-soluble aerosols measured during Easter Bush campaign, calculated from hourly data. Number of measurements (N) for each compound is also shown.

| (2.4 m) | LOD ng m$^{-3}$ | $\mu_A$ µg m$^{-3}$ | $\mu_M$ µg m$^{-3}$ | Min µg m$^{-3}$ | Max µg m$^{-3}$ | $\sigma_A$ µg m$^{-3}$ | N |
|---|---|---|---|---|---|---|---|
| $NH_4^+$ | 190 | 0.74 | 0.64 | <LOD | 2.33 | 0.43 | 580 |
| $Cl^-$ | 15 | 0.91 | 0.36 | <LOD | 7.88 | 1.31 | 515 |
| $NO_2^-$ | 17 | 0.02 | 0.02 | <LOD | 0.05 | 0.01 | 373 |
| $NO_3^-$ | 47 | 1.53 | 1.32 | <LOD | 6.27 | 1.18 | 538 |
| $SO_4^{2-}$ | 109 | 1.29 | 1.22 | <LOD | 6.26 | 0.83 | 540 |
| $NH_3$ | 172 | 1.48 | 1.15 | <LOD | 13.79 | 1.5 | 602 |
| HCl | 67 | 0.2 | 0.15 | <LOD | 1.4 | 0.18 | 544 |
| HONO | 30 | 0.04 | 0.04 | <LOD | 0.12 | 0.02 | 410 |
| $HNO_3$ | 97 | 0.19 | 0.16 | <LOD | 0.68 | 0.12 | 509 |
| $SO_2$ | 120 | 0.24 | 0.18 | <LOD | 1.48 | 0.21 | 480 |

**Table 2: Mean (µA), median (µM), minimum and maximum vales for flux, deposition velocity ($V_d$), maximum deposition velocity ($V_{max}$), and canopy resistances ($R_c$) for trace gases measured during Easter Bush campaign, based on hourly values. Also shown are the median relative standard error ($\sigma_F$), the flux limit of detection ($F_{LOD}$) evaluated for typical conditions (median $u_*$ and median concentration) as well as the fraction of the hourly flux value that exceed the flux detection limit evaluated for that hour ($f_{LOD}$).**

| | | $NH_3$ | HCl | HONO | $HNO_3$ | $SO_2$ |
|---|---|---|---|---|---|---|
| **Flux (ng m$^{-2}$ s$^{-1}$)** | $\mu_A$ | 15.24 | -3.51 | -0.3 | -2.66 | -3.04 |
| | $\mu_M$ | 5.65 | -1.98 | -0.29 | -1.99 | -1.68 |
| | Min | -324.5 | -61.24 | -2.46 | -18.57 | -35.57 |
| | Max | 1460 | -0.03 | 4.92 | 0.82 | -0.03 |
| | No. of measurements | 577 | 506 | 384 | 500 | 465 |
| | $s_F$ (%) | 32 | 58 | 56 | 42 | 67 |
| | $F_{LOD}$ | 1.28 | 0.75 | 0.18 | 0.89 | 0.97 |
| | $f_{LOD}$ (%) | 94 | 84 | 78 | 87 | 89 |
| **$V_d$ (mm s$^{-1}$)** | $\mu_A$ | -8.99 | 15.1 | 8.8 | 13.61 | 11.69 |
| | $\mu_M$ | -6.1 | 14.49 | 7.69 | 12.87 | 10.00 |
| | Min | -215.3 | 0.01 | -55.6 | -4.72 | 0.34 |
| | Max | 92.90 | 52.83 | 59.81 | 56.78 | 55.38 |
| **$V_{max}$ (mm s$^{-1}$)** | $\mu_A$ | 19.46 | 15.33 | 14.12 | 13.91 | 14.22 |
| | $\mu_M$ | 18.8 | 15.26 | 13.99 | 13.75 | 14.02 |
| | Min | 1.7 | 0.04 | 0.04 | 0.04 | 0.45 |
| | Max | 57 | 40.41 | 36.99 | 36.93 | 36.99 |
| **$R_c$ (s m$^{-1}$)** | $\mu_A$ | 0 | 33.75 | 331.5 | 23.29 | 49.20 |
| | $\mu_M$ | 0 | 1.82 | 13.07 | 5.71 | 27.61 |

**Table 3: Mean (μA), median (μM), minimum and maximum vales for flux and deposition velocity (V$_d$) for water soluble aerosols measured during Easter Bush campaign. Also shown are the median relative standard error (σ$_F$), the flux limit of detection (F$_{LOD}$) evaluated for typical conditions (median $u_*$ and median concentration) as well as the fraction of the hourly flux value that exceed the flux detection limit evaluated for that hour (f$_{LOD}$).**

|  |  | NH$_4^+$ | Cl$^-$ | NO$_3^-$ | SO$_4^{2-}$ |
|---|---|---|---|---|---|
| **Flux (ng m$^{-2}$ s$^{-1}$)** | μ$_A$ | -3.55 | -4 | -3.34 | -3.56 |
|  | μ$_M$ | -2.97 | -1.11 | -1.76 | -2.19 |
|  | Min | -42.23 | -60.04 | -89.32 | -59.69 |
|  | Max | 18.15 | -1.06 | 31.84 | -0.95 |
|  | No. of measurements | 224 | 484 | 477 | 482 |
|  | σ$_F$ (%) | 58 | 41 | 48 | 45 |
|  | F$_{LOD}$ | 2.21 | 0.85 | 1.28 | 1.78 |
|  | f$_{LOD}$ (%) | 91 | 81 | 84 | 87 |
| **V$_d$ (mm s$^{-1}$)** | μ$_A$ | 0.93 | 3.65 | 1.97 | 1.89 |
|  | μ$_M$ | 0.37 | 3.14 | 1.52 | 1.45 |
|  | Min | -0.04 | -0.92 | -9.43 | -2.48 |
|  | Max | 7.57 | 21.26 | 9.8 | 9.53 |