# Peer review of "Surface—atmosphere exchange of inorganic water—soluble gases and associated ions in bulk aerosol above agricultural grassland pre— and post—fertilisation"

_Atmospheric Chemistry and Physics, 2018_

## Referee Comment (RC1) · Anonymous Referee #2 · 6 Aug 2018

After thorough reading the manuscript I come to the conclusion that it does not meet the standards of atmospheric chemistry and physics and has to be rejected. My rating is based on several points:

Title and abstract promise measurements, findings, and discussions which are not given. Title and abstract are very broadly formulated, while the paper itself lacks of focus.

Substantial supportive measurements are lacking (e.g. aerosol size distribution or even size resolved chemical analysis of the aerosol).

[Figure]

The text is not structured clearly and way too long.

Some sections contradict each other.

When studying reactive trace gas exchange fluxes, possible flux divergence needs to be addressed. The typical sources for flux divergences are introduced in the introduction but not analyzed and discussed in the paper.

There are several indications for flux divergence in the results. Nonetheless the authors calculate a 'flux' from the measured gradients and even derive a canopy resistance.

Flux limits of detection are explained in the material and method section, but no results are given. Small and bidirectional fluxes most probably were within the detection limit.

At a well-studied site like Easter Bush there should be more information on aerosol chemistry than just the GRAEGOR measurements. The comparison with MARGA results (measured at a distance of 12 km) itself plus the very rough aerosol size analysis is not sufficient.

Furthermore the supportive measurements of the MARGA are not described in the cor-responding section. Nor are the NO2 measurements.

In the comparison of GRAEGOR measurements with LOPAP and QCL measurements discussion is mixed with contents that should better be placed in the material and method and/or the results section. Data for both comparison lack in number, range and supportive measurements, which would help to understand agreement and dis-agreement.

The presented measurements and results do not lead to the presented conclusions.

Conclusions remain speculative, unfounded and airy.

**Conclusion**

The discussion paper does not keep up to the promising title and abstract. The data basis does not appear to bring sufficient material to a paper on its own. Maybe the data can be presented as supportive data in another paper, such as the cited Di Marco et al. one on HONO fluxes.

---

## Author Comment (AC1) · 15 Aug 2018

We thank the reviewer for their time spent reading our paper and providing comments.

Clearly we have failed to convince this reviewer of the merits of our dataset and of our interpretations of this dataset. Below we provide responses to the reviewer's criticisms on a point-by-point basis (the reviewer's comments are provided in quotation marks, and are numbered), but first we wish to reiterate the aims of our study and the relevancy of its conclusions.

The aim of this work was to determine, at hourly resolution for a month, the concen-

trations and fluxes of water-soluble trace gases and their associated particle-phase ionic counterparts as measured by the Gradient of Aerosols and Gases Online Registration (GRAEGOR) at two heights over agricultural grassland. The vertical fluxes of these species were calculated using the modified aerodynamic gradient method and co-located micrometeorological measurements. Simultaneous time-resolved fluxes of these atmospheric components have not been widely determined because of the sophistication of the instrumentation required to do so. We carefully considered issues of limits of detection and flux divergence.

A further aim of this study was to discuss any change in flux of reactive nitrogen species after an inorganic fertilizer application to the grassland part way through the measurement period. Such a change was observed.

It was not an aim of this study to measure fluxes of particles, total or size-resolved. The use of the word 'aerosol' without qualification in the title of this paper may have unintentionally raised expectation that full particle flux characterisation is included – we expand on this remark, and suggest a refinement to the title, further below.

Our paper presents bulk deposition velocities of the water-soluble aerosol-phase ions $Cl^-$, $NO_3^-$ and $SO_4^{2-}$ which is important knowledge for deposition models. In examining the deposition velocities we derived we hypothesised a relationship with the proportion of fine and coarse particulate matter. We were able to demonstrate this association by plotting hourly deposition velocity as a function of the temporal PM2.5/PM10 ratio measured nearby. We emphasise throughout our paper that use of this ratio is a proxy only for particle size. This reviewer might have liked to see additional measurements that would have allowed a more quantitative analysis. However, to our knowledge, this is the first study that through composition-resolved bulk flux measurements confirms the enhancement in the bulk deposition velocity of aerosol constituents partly contained in the coarse fraction. Direct particle number flux measurements in the coarse fraction are notoriously difficult due to the limited counting statistics. The only ambient measurements available are from fog droplet deposition (see reviews, e.g. by Pryor

et al., 2008), based on the (non-validated) assumption that aerosols and fog droplets interact with vegetation in similar manner.

A final aim of our work was to present a novel intercomparison between measurements of nitrous acid as gathered by two wet-chemistry instruments.

We maintain that fundamentally the depth of our dataset, its presentation and the conclusions it supports are appropriate for publication in ACP.

1. "After thorough reading the manuscript I come to the conclusion that it does not meet the standards of atmospheric chemistry and physics and has to be rejected. My rating is based on several points: Title and abstract promise measurements, findings, and discussions which are not given. Title and abstract are very broadly formulated, while the paper itself lacks of focus."

We do not agree that the abstract is broadly formulated or promises measurements and discussions that are not subsequently presented. The method by which the trace gases and associated aerosol counterparts were measured is specifically mentioned in the abstract, as is the method by which flux was calculated. All findings referred to in the abstract are directly referred to in the results section, and the discussion points in the abstract are likewise presented in the discussion sections of the paper. We acknowledge, however, that the following wording in the abstract "direct evidence of a size-dependence of aerosol deposition velocity" is overstated, since aerosol size distributions were not directly measured at the site; but the conclusion itself is supported by use of a proxy measurement of aerosol size, which we stress is a proxy measurement throughout the text, including later in this same sentence in the abstract. We will re-word this sentence in the revised paper to remove the phrasing "direct evidence." We will also add to the abstract the location of the study. Other than that we do not see anything in the abstract which is not supported by data discussed in the paper.

We do acknowledge that the title of the paper is potentially too broad in that it does not specifically qualify that the aspect of aerosol fluxes measured are the water-soluble

ionic counterparts of the measured trace gases not full particle distribution fluxes. However, we believe that the exact scope of the aerosol-phase measurements made in our work is readily clear from the abstract and the main text. To avoid ambiguity we will extend the title of our paper to "Surface-atmosphere exchange of inorganic water-soluble gases and associated ions in bulk aerosol above agricultural grassland pre- and post-fertilisation."

2. "Substantial supportive measurements are lacking (e.g. aerosol size distribution or even size resolved chemical analysis of the aerosol)."

We believe that the origin of this comment may relate to the issue raised above concerning the abstract, which mentions aerosol size, but which is clarified within the abstract and throughout the text as being a proxy measurement of fine to coarse particle ratio. As indicated above, we will amend the phrasing on this in the abstract and in the title. Measurement of aerosol size distributions and full size-dependent chemical composition was not a component of our study, and we only base our conclusions on data that were available to us. We state in Section 4.2.2 that we did not make size-resolved aerosol measurements: "Although measurements of particle size were not made during this campaign, measurements of aerosol species (including $Cl^-$ and $SO_4^{2-}$) in the PM2.5 and PM10 size fractions were taken by a two-channel Monitor for Aerosols and Gases in Ambient Air (MARGA, Applikon B.V, The Netherlands) instrument located at Auchencorth Moss, 12 km south west of Easter Bush…As proxy for a particle size measurement, the proportion of PM2.5 to PM10 was used, with a lower proportion of PM2.5 indicating a greater proportion of coarse aerosol, and a corresponding larger deposition velocity based on process-orientated modelling".

We also state in our Conclusions section that "Future measurements of aerosol deposition velocities should aim to investigate the effect of particle size upon deposition velocity, using a more robust measurement of particle size than used here".

We believe that re-emphasis of the proxy nature of aerosol measurements in the abstract will clarify this matter.

3. "The text is not structured clearly and way too long."

The reviewer has not amplified on aspects of the paper that they feel are not structured clearly. We have followed the standard structure of presenting primary results and discussions in separate sections. The Discussion includes secondary analysis of the results data, and/or other data brought in, where this supports the points we wish to draw out at a particular place in the Discussion.

4. "Some sections contradict each other."

We cannot provide a response to this since the reviewer has not specified where they believe there is contradiction. We do not see a contradiction in what we present.

5. "When studying reactive trace gas exchange fluxes, possible flux divergence needs to be addressed. The typical sources for flux divergences are introduced in the introduction but not analyzed and discussed in the paper. There are several indications for flux divergence in the results. Nonetheless the authors calculate a 'flux' from the measured gradients and even derive a canopy resistance."

We agree that the period of measurement includes periods of flux divergence due to changes in the gas-aerosol partitioning. However, this has been explicitly addressed throughout the manuscript. At the end of Section 2.3.3 we state that we initially process the data ignoring chemistry and discuss the validity of this assumption later. Divergence of Vd(HNO3) (as calculated neglecting chemistry) from Vmax has often been taken as an indicator of the importance of flux divergence as have very large deposition velocities of NH4+. Thus, in Section 4.2.3 it is shown that the influence of chemistry is within the measurement uncertainty, except for the period after fertilisation. To overcome this problem, this period is treated separately by calculating the conservative tracers of total-ammonium and total-nitrate, as per our text: "It should be noted that during this period the aerodynamic gradient method does not derive accurate fluxes

because the condition of flux conservation is not met... By contrast, fluxes of total ammonium and total nitrate would be conserved, as the effect of gas-particle interactions are not considered, and their assessment provides additional information on the processes occurring during periods when fluxes are not conserved with height." Our conclusions based on the behaviour of ammonia, ammonium nitrate and nitric acid fluxes are grounded in this analysis.

We have also taken care to exclude the period of flux divergence from the statistical analysis of canopy resistance and deposition velocity for the entirety of the campaign. This latter point was perhaps not clear in the paper and we will add text to stress this exclusion.

It is also worth noting in this context, that previous analyses indicate that size-segregated particle number fluxes (e.g. Nemitz and Sutton, 2004) and even total particle number fluxes (Nemitz et al., 2009) can be highly perturbed by gas-aerosol partitioning, an artefact that is usually completely ignored. This implies that eddy-covariance particle number fluxes do not have a methodological advantage over the bulk composition gradient flux measurements presented here.

6. "Flux limits of detection are explained in the material and method section, but no results are given. Small and bidirectional fluxes most probably were within the detection limit."

The flux limits of detection for all trace gas and aerosol species measured will be added to Tables 4 and 5, respectively, of the manuscript. The bidirectional fluxes of HONO and NH3 that were observed throughout the campaign – including the period before fertilisation – remained above the flux detection limit calculated. This point will also be added to the text.

7. "At a well-studied site like Easter Bush there should be more information on aerosol chemistry than just the GRAEGOR measurements. The comparison with MARGA results (measured at a distance of 12 km) itself plus the very rough aerosol size analysis

is not sufficient."

As outlined in Section 2.1., Easter Bush periodically hosts campaigns but it is not a site which hosts a suite of long-term measurements of aerosol chemistry. Aerosol size measurements, if they had been available, would have been used. Standard continuous measurements of size distribution would not have greatly aided this campaign: scanning mobility particle sizers (SMPS) and Aerosol Mass Spectrometers (AMS) measure in the sub-micron fraction and the AMS only non-refractory aerosol, whilst analysis focuses on the influence of the coarse fraction and refractory material on deposition velocity. Whilst impactor measurements of aerosol size segregated ion composition would have been helpful, they were not available and indeed are not a routine measurement even at Supersites.

8. "Furthermore the supportive measurements of the MARGA are not described in the corresponding section. Nor are the NO2 measurements."

We thank the reviewer for pointing out these omissions. A full description of the MARGA set-up and operation at the Auchencorth site is available in (Twigg et al, 2015). The processed and ratified MARGA data are publicly available online at - https://uk-air.defra.gov.uk/data/data_selector -from which concentrations of any of the species measured by the MARGA can be selected. We will add the references and online resources to the paper.

We also acknowledge that whilst measurements of NO2 are mentioned in Section 4.4.1 the details of this measurement are not included. The NO2 concentrations were determined by chemiluminescence analyser operated to standard UK national network protocols. Details will be added to the revised paper.

9. "In the comparison of GRAEGOR measurements with LOPAP and QCL measurements discussion is mixed with contents that should better be placed in the material and method and/or the results section. Data for both comparison lack in number, range and supportive measurements, which would help to understand agreement and disagreement."

During the preparation of this paper the authors discussed the best placements within the paper of the material on the LOPAP and QCL comparisons. We consider that the comparison between GRAEGOR and LOPAP, and GRAEGOR and QCL, is best placed in the Discussion Section as this was not the motivation of the study and we felt that it would become disjointed if material was split between several sections. If reviewers collectively feel strongly on this point we are happy to rearrange the sections.

With respect to the comment on amount of intercomparison data, for the comparison between the QCL and the GRAEGOR there is a limited (n = 72) number of measurements. This is considered in the discussion of the comparison (Section 4.4.2), where we mention that lack of measurements restricted comparison to only concentrations of ammonia. On the other hand, the comparison between the LOPAP and the GRAEGOR spans 6 days (n = 148). We are mindful of the ovelty of the comparisons – the first that we are aware of to be presented for publication – and we believe the data and our observations on it are worth inclusion in the paper.

10. "The presented measurements and results do not lead to the presented conclusions... Conclusions remain speculative, unfounded and airy... The discussion paper does not keep up to the promising title and abstract. The data basis does not appear to bring sufficient material to a paper on its own. Maybe the data can be presented as supportive data in another paper, such as the cited Di Marco et al. one on HONO fluxes."

These final comments from the reviewer essentially re-iterate the same concerns the reviewer has expressed in their earlier comments, and to which we have responded. In summary, we present one full month of hourly-resolved multi-species flux measurements, and throughout the results and discussion section, present carefully considered findings and conclusions which we have made based on our understanding of atmosphere science, the scientific literature, and analysis of the data gathered. We will

Interactive
comment

make modifications to the title and the abstract to avoid any interpretation that this study included size-resolved aerosol flux measurements, plus the other modifications indicated above.

References - Pryor, S. C.; Gallagher, M. W.; Sievering, H.; Larsen, S. E.; Barthelmie, R. J.; Birsan, F.; Nemitz, E.; Rinne, J.; Kulmala, M.; Groenholm, T.; Taipale, R.; Vesala, T., A review of measurement and modelling results of paricle atmosphere-surface exchange. Tellus B 2008, 60, 42-75.

Nemitz, E.; Sutton, M. A., Gas-particle interactions above a Dutch heathland: III. Modelling the influence of the NH3-HNO3-NH4NO3 equilibrium on size-segregated particle fluxes. Atmospheric Chemistry and Physics 2004, 4, 1025-1045.

Nemitz, E.; Dorsey, J. R.; Flynn, M.; Gallagher, M. W.; Hensen, A.; Erisman, J. W.; Owen, S.; Daemmgen, U.; Sutton, M. A., Aerosol fluxes and particle growth above grassland following the application of NH4NO3 fertilizers. Biogeosciences 2009, 6, 1627-1645.

Twigg; M. M.; Di Marco; C. F.; Leeson; S.; Van Dijk; N.; Jones; M. R.; Leith; I. D.; Morrison; E.; Coyle; M.; Proost; R.; Peeters; A. N. M.; Lemon; E.; Frelink; T.; Braban; C. F.; Nemitz; E.; Cape; J. N., Water soluble aerosols and gases at a UK background site – Part 1: Controls of PM2.5 and PM10 aerosol composition, Atmospheric Chemistry and Physics 2015, 15; 8131–8145; doi:10.5194/acp-15-8131-2015; 2015.
* * *

---

## Referee Comment (RC2) · Anonymous Referee #1 · 17 Aug 2018

Review of Surface–atmosphere exchange of water-soluble gases and aerosols above agricultural grassland pre- and post-fertilisation by Robbie Ramsay et al. I find this manuscript somewhat difficult to evaluate, because while on the one hand there is a need for improved understanding of (and model tools for estimating) volatilization of reactive species after fertilization, on the other-hand does this manuscript offer a real advancement of the current state of knowledge? I think not in its current state and FEAR that the measurements are not actually to a standard necessary to address the key points that authors wish to resolve. My key concerns are; 1) As illustrated by many of the figures using gradients to determine fluxes is extremely difficult (see the concentration plots in Fig 3 & 4). Indeed direct flux measurements are also very difficult! Thus

although the fluxes shown in Figure 7 and Figure 9 are presented without error bars I suspect the error bars are in fact VERY large. This is not a new problem and is certainly not unique to these authors or this study. BUT Figure 10 actually tells an important part of the story as does in Figure 14 . . .that the concentrations are themselves rather uncertain. 2) Fundamentally is GRAEGOR 'fit for purpose'? Some basic statistics could be brought into play to consider what fraction of flux periods (of each of the considered species) exceeded the uncertainty bounds FOR each individual measurement. The authors describe some efforts at determining uncertainty in concentrations and fluxes but they do not appear to be applied and the description is quiet vague – and associated with statements I find it hard to comprehend; 'Uncertainties for the trace gases and water–soluble aerosols measured calculated by error propagation ranged from 8% - 18% ($3\sigma$) throughout the campaign, varying primarily due to fluctuations in the measured flow rate and analysed concentration of the internal Br standard.' Does this really mean ALL species for ALL hours had an uncertainty of 8-18% of the measured concentration? '$\sigma u^*$ was estimated at 12% median, which, in combination with $\sigma \Delta c$, was used to calculated $\sigma F$' – I can't see uncertainties are presented. . . 'While most exceedances fall within the uncertainty range of the measurement' . . . How many do not? And why? 3) Addressing point 2) and doing so in a manner that actually uses uncertainties for EACH measurement not for the sample as a whole would be useful in contextualizing the flux estimates and allowing the authors to determine if the 'good enough' threshold is achieved. 4) I think Figure 12 is partly a response to particle size but since no data on particle size were provided is it also a story of large measurement uncertainty? 5) The manuscript title implies a focus on fluxes ("Surface–atmosphere exchange of water-soluble gases and aerosols above agricultural grassland pre- and post-fertilisation") why are so many of the figures and so much of the text about concentrations and/or the ion balance in the aerosols? Perhaps this reflects the author's own assessment that GRAEGOR is not adequate to derive robust fluxes. If it is not then the manuscript really does not meet the bar for ACP but parts of the research could be published in another forum. 6) With only a single fertilization event I wonder

how generalizable this is? If a data set could be developed that comprises many fertilization events it may be possible to extract a signal, but at the moment the S2N ratio is very low. 7) I think IF a numerical model (that accounts for flux divergence) could be brought into the research it would be very useful in trying to extract more information and provide greater insights. As it stands I did not find it compelling and thus the conclusions seem to really over-state what is shown in the manuscript. It's a minor point given the above but although the manuscript is quite lengthy, I did not find all the details of the measurements.

---

## Author Comment (AC2) · 4 Sep 2018

**acp-2018-603: Surface–atmosphere exchange of water-soluble gases and aerosols above agricultural grassland pre- and post-fertilisation**
**by Ramsay, R. et al.**

**Response to Anonymous Referee #1**

We thank the reviewer for their time in reviewing our paper and providing comments. Before addressing the numbered points made by this reviewer individually, we would like to respond to the reviewer's overall question "does this manuscript offer a real advancement of the current state of knowledge?" We believe our paper provides novelty and advancement for the following reasons:

Firstly, there is currently a lack of data for simultaneous time-resolved fluxes of trace inorganic gases and associated aerosol counterparts, particularly for the reactive nitrogen species $NH_3/NH_4^+$ and $HNO_3/NO_3^-$. Our paper presents flux data for these species over agricultural grassland at hourly resolution for one month at high precision and with appropriate consideration of the uncertainties in the flux values. This dataset also includes fluxes for trace gas and aerosol species during a period of flux divergence post-application of urea fertiliser. By careful consideration of the issues present in analysing fluxes during periods of flux divergence, we present a robust dataset which considers total nitrate and total ammonium fluxes during this period and discusses the changes in flux behaviour post-fertiliser application. Furthermore, it provides strong field evidence of a ground source of both HONO and $HNO_3$ after fertilisation.

Secondly, this paper presents bulk deposition velocities for particulate $Cl^-$, $NO_3^-$ and $SO_4^{2-}$, which are themselves important values for deposition modelling. From observation of these deposition velocities, it was hypothesized there was a link between deposition velocity and the proportion of fine to coarse aerosol. Using the ratio of $PM_{2.5}/PM_{10}$, as measured by a nearby instrument, we were able to demonstrate this association. While this proportion acts as a proxy measurement for particle size measurements, we believe this is novel evidence for demonstrating a link between enhancement of aerosol deposition velocity and proportion of particles contained in the coarse fraction.

Thirdly, we believe we present the first intercomparison data for nitrous acid measurements made by the Gradient of Aerosols and Gases Online Registration (GRAEGOR) and by the Long Path Absorption Photometer (LOPAP), which not only compares concentrations, but also gradients. We also present an intercomparison of ammonia measurements made by the GRAEGOR and by a Quantum Cascade Laser (QCL).

We now respond to the individual points raised in the review, with the reviewer's comments presented in blue, italicized font.

*1) "As illustrated by many of the figures using gradients to determine fluxes is extremely difficult (see the concentration plots in Fig 3 & 4). Indeed direct flux measurements are also very difficult! Thus although the fluxes shown in Figure 7 and Figure 9 are presented without error bars I suspect the error bars are in fact VERY large. This is not a new problem and is certainly not unique to these authors or this study. BUT Figure 10 actually tells an important part of the story as does in Figure 14…that the concentrations are themselves rather uncertain."*

We agree that it would be helpful to include error bars on the time series for fluxes and will add these to Figures 7, 9 and 13 in the revised manuscript. We have included in this response a revised version of Figure 7 – the time series of trace gas fluxes - which includes error bars as an example. We will also add a summary of median flux error values to Tables 4 and 5.

[Figure]

Figure 7: Time series of hourly trace gas fluxes measured during the Easter Bush campaign. Results smoothed using a 5-point moving point average. The fertilisation period was 08:00 – 09:00 on 13th June, and is highlighted in green. Flux uncertainties for each trace gas are included as error bars.

Figure 10 (which compares measurements of $NH_3$ taken by the GRAEGOR to that by a QCL system) and Figure 14 (similarly comparing measurements of HONO taken by the GRAEGOR to that by a LOPAP) are intercomparison studies between instruments. These comparisons do not reflect the uncertainty of GRAEGOR concentration measurements, only that there exists a difference between measurement techniques that should be accounted for when considering measurements of concentration. Crucially, the difference in measurements between two different systems does not directly impact on the error in the concentration gradient of the GRAEGOR, which is the critical part in calculating flux values. The two GRAEGOR detector boxes share the same analytical system and therefore uncertainties in the concentrations at the two heights are not independent, and the error on the gradient is significantly smaller than the combined error between the two concentrations.

It is our view that the flux uncertainties in this study are not "very large" in respect to previously published studies using the GRAEGOR. The median flux error for $NH_3$, for example, was 32%, which is similar to values obtained for measurements made over grassland using the GRAEGOR by Wolff et al. (2010) and Thomas et al. (2009). By inclusion of median flux error values in Tables 4 and 5, we anticipate we can satisfy readers that our flux errors are in the range expected for use of this instrument.

*2) "Fundamentally is GRAEGOR 'fit for purpose'? Some basic statistics could be brought into play to consider what fraction of flux periods (of each of the considered species) exceeded the uncertainty bounds FOR each individual measurement."*

The reviewer raises an important point that was considered during flux calculations, but which was not included in the manuscript. As outlined by Thomas et al. (2009), it is possible to calculate the minimum flux that the GRAEGOR can measure for each species, effectively providing a limit of detection for flux measurements. Fluxes presented have been filtered using these values, but discussion of their development, beyond mentioning that fluxes were filtered according to these values was not included, nor were they included in Tables 4 and 5. We will include a brief discussion of calculating this value in the Methods section of the revised manuscript, which, together with inclusion of minimum detectable flux values for each species in Tables 4 and 5, should resolve this issue.

It is important to emphasize that the capability of the GRAEGOR to measure fluxes is dependent upon its ability to measure concentration differences with sufficient precision. As documented in the manuscript, a side-by-side comparison of the GRAEGOR sample boxes was used to develop a linear regression profile, from which – after correction – the residuals were used to determine the precision of the concentration measurements. As we state in our manuscript: "From the results obtained, it was found that for the gases $NH_3$, HCl, HONO, $HNO_3$ and $SO_2$ that deviation from the 1:1 fit resulted in a precision of measurements <4% ($3\sigma$). For the aerosol species $Cl^-$, $NO_3^-$ and $SO_4^{2-}$, precision was calculated as <8% ($3\sigma$), while for $NH_4^+$ was calculated as <9% ($3\sigma$)." These precision values are in line with those previously calculated by those using the GRAEGOR to measure fluxes (Thomas et al., 2009; Wolff et al., 2010; Twigg et al., 2011), and we maintain that these are sufficient precision values to resolve the vertical concentration gradients necessary for flux calculations.

The GRAEGOR shares its principle of operation and many components with other instrumentation that has routinely been used for gradient flux measurements, such as the AMANDA/GRAHAM for $NH_3$ (same denuder but based on selective membrane / conductivity; Erisman and Wyers, 1993; Flechard and Fowler, 1998; Milford et al., 2001, 2009; Wichink Kruit et al., 2007; Neirynck et al., 2008) and the MARGA-based gradient system of Rumsey et al. (2016). Considering the similar architecture of the GRAEGOR to these instruments which have been successfully used to measure gradient fluxes, the statement that the GRAEGOR would not be fit for purpose is therefore surprising.

Finally, the good agreement between expected and measured deposition velocities for $HNO_3$ and HCl may be taken as independent evidence (though not proof) of the high quality of the measured fluxes.

*"The authors describe some efforts at determining uncertainty in concentrations and fluxes but they do not appear to be applied"*

As described in responses above, we will resolve issues of clarity surrounding the uncertainty measurements by revising quoted figures in the text so that they include their error values. We will also include error bars in figures where appropriate, and revise Tables 4 and 5 to include further details on error measurements.

*"…the description is quiet[sic] vague and associated with statements I find it hard to comprehend; 'Uncertainties for the trace gases and water–soluble aerosols measured calculated by error propagation ranged from 8% - 18% (3σ) throughout the campaign, varying primarily due to fluctuations in the measured flow rate and analysed concentration of the internal Br standard.' Does this really mean ALL species for ALL hours had an uncertainty of 8-18% of the measured concentration?"*

We agree with the reviewer that the wording of this section lacks clarity. The inclusion of the determination of instrument error was part of a development to determine flux error. However, as detailed in Section 2.3.4 "Alternatively, the full random error can be characterised experimentally…", the full random error was found through side by side measurements of the GRAEGOR sample boxes. This vagueness is compounded by an error in stating that concentration errors were developed from the calculation of instrument error, rather than from the side-by-side measurements. We will clarify this issue in the revised manuscript.

*'σu\* was estimated at 12% median, which, in combination with $\sigma_{\Delta c}$, was used to calculated σF' – I can't see uncertainties are presented… 'While most exceedances fall within the uncertainty range of the measurement' How many do not? And why?*
*3) Addressing point 2) and doing so in a manner that actually uses uncertainties for EACH measurement not for the sample as a whole would be useful in contextualizing the flux estimates and allowing the authors to determine if the 'good enough' threshold is achieved."*

As discussed above, we shall resolve this issue through the inclusion of the necessary values in the text.

*4) "I think Figure 12 is partly a response to particle size but since no data on particle size were provided is it also a story of large measurement uncertainty?"*

In our discussion of the investigation into the dependence of measured deposition velocity ($V_d$) on particle size, we repeatedly emphasize that our value for particle size is a proxy (the ratio $PM_{2.5}/PM_{10}$ as measured by an instrument nearby). As stated in the text – "Although measurements of particle size were not made during this campaign, measurements of aerosol species (including $Cl^-$ and $SO_4^{2-}$) in the $PM_{2.5}$ and $PM_{10}$ size fractions were taken by a two-channel Monitor for Aerosols and Gases in Ambient Air (MARGA, Applikon B.V, The Netherlands) instrument located at Auchencorth Moss, 12 km south west of Easter Bush…As proxy for a particle size measurement, the proportion of $PM_{2.5}$ to $PM_{10}$ was used, with a lower proportion of $PM_{2.5}$ indicating a greater proportion of coarse aerosol, and a corresponding larger deposition velocity based on process-orientated modelling". Particle size measurements were not available. As summarised at the start of this response document, the proxy measurement was used to investigate a hypothesis that was developed from observations of aerosol $V_d$ values. Our use of such a proxy measurement is not related to measurement uncertainty. Figure 12 shows strong and statistically significant relationships.

The scatter indicated may reflect measurement uncertainty, but could equally reflect limitations in a concentration ratio from a nearby site to describe the full size-distribution at our measurement site, the additional effect of atmospheric stability on $V_d/u_*$ (e.g. Wesely et al., 1985) or a number of additional processes (e.g. surface wetness). We anticipate that the reviewer's concern will be resolved by further clarifying the proxy nature of the aerosol size measurement in our manuscript.

*5) "The manuscript title implies a focus on fluxes ("Surface–atmosphere exchange of water-soluble gases and aerosols above agricultural grassland pre- and post-fertilisation") why are so many of the figures and so much of the text about concentrations and/or the ion balance in the aerosols?"*

We believe it is necessary to include figures and text discussing concentrations as a precursor to discussion of fluxes. It is also important to present these findings for the discussions relating to (i) the deposition velocities (which makes references to elevated periods of $Cl^-$ concentrations that are visually apparent in Figure 3), (ii) the HONO fluxes (the presence of HONO concentrations above the detection limit suggests a day time source for HONO; this is then linked to discussion of HONO fluxes), and (iii) the instrument intercomparison studies (which compare concentrations).

The inclusion of a brief discussion of ionic balance, with an accompanying figure, was necessary to discuss the development of the hypothesis of aerosol bulk deposition velocities being enhanced by particles in the coarse fraction. As mentioned in the text – "[the ionic balance study]…suggests a deficit of $NH_4^+$, suggesting that some of the $NO_3^-$ and/or $SO_4^{2-}$ was balanced by ions other than $NH_4^+$. A likely candidate is $Na^+$: some of the $SO_4^{2-}$ is likely to have represented sea-salt $SO_4^{2-}$ and some $NaNO_3$ is formed by reaction of $NaCl$ with $HNO_3$." This is then followed by discussion of atmospheric chemical processes that would give rise to the formation of these coarse particles, providing the framework for the eventual discussion of aerosol $V_d$ and particle size proxy. Without the inclusion of this section, we believe that it would harm the coherence of a novel discussion point in the paper.

In conclusion, we believe that the discussion of concentration data is a necessary part of further discussion surrounding surface-atmosphere exchange. We could alternatively have called the manuscript "Concentrations and surface/atmosphere exchange fluxes of water-soluble …", but we feel that beyond making the title even more cumbersome than it is already, this would not add any more information.

*6) "With only a single fertilization event I wonder how generalizable this is? If a data set could be developed that comprises many fertilization events it may be possible to extract a signal, but at the moment the S2N ratio is very low."*

Our paper provides rare field evidence of a ground source of both HONO and $HNO_3$ following fertilisation, corroborating previous evidence of Sutton et al. (1998) and Twigg et al. (2011). These results are quite remarkable because it is certainly not clear why the application of fertiliser, characterised by a high pH, should result in $HNO_3$ release. We do not suggest in the paper that our results should be generalized to fit all grassland fertilization events. However, the observations may well trigger follow-on laboratory process studies of these fertiliser emissions of HONO and $HNO_3$.

Here, we aimed only to observe the fluxes of trace gases and aerosols, particularly reactive nitrogen species, pre- and post- fertilisation of a grassland site with urea fertiliser, and to discuss any observed changes. We believe that through our use of the chemical conservative tracers total-nitrate and total-ammonium, we have accounted for the period of flux divergence while drawing relevant conclusions about the behaviour of reactive nitrogen species postfertilisation. These results add to the literature on fluxes of reactive nitrogen above fertilised grassland, much of which has also described study of only one fertilisation event.

*7) "I think IF a numerical model (that accounts for flux divergence) could be brought into the research it would be very useful in trying to extract more information and provide greater insights. As it stands I did not find it compelling and thus the conclusions seem to really over-state what is shown in the manuscript."*

The senior author of this publication is indeed a global leader in the 1D modelling of the $NH_3$-$HNO_3$-$NH_4NO_3$ interaction (e.g. Nemitz et al., 1998; Nemitz et al., 2000; Nemitz et al., 2009; Ryder et al., 2016). A general thread running through these model studies has been that the models are able to explore the observations qualitatively, but that it is difficult to constrain the model sufficiently to provide fully quantitative results. For example, a fully quantitative model run would require treatment and measurement information of the aerosol composition as a function of size, including any potential external mixing. Such model application is, however, well beyond the scope of this paper or the comprehensiveness of the dataset.

We do not believe that we have overstated our conclusions, as all conclusions cited in the abstract and conclusion sections are argued through the results and discussion section appropriately. We maintain that the data is robust, but we shall emphasize in the revised manuscript why we believe that to be the case, providing more information on flux errors and minimum detectable fluxes, as well as clarifying the issue of concentration measurement errors. We will also reword one conclusion in the abstract, which currently reads "providing direct evidence of a size-dependence of aerosol deposition velocity for aerosol chemical compounds" to remove the phrase "direct evidence", which we hope in combination with emphasizing the proxy nature of aerosol size measurements should clarify our hypothesis regarding aerosol $V_d$ and aerosol size. In conclusion, we belief that the work presented in the paper remains suitable for publication by ACP, with the above-discussed amendments to highlight the robustness of our dataset.

*"It's a minor point given the above but although the manuscript is quite lengthy, I did not find all the details of the measurements."*

We think the reviewer may be referring here to some lack of information on the source of the $NO_2$ concentrations and on the MARGA instrument measurements, to which Reviewer #2 also referred. The $NO_2$ concentrations were determined by chemiluminescence analyser operated to standard UK national network protocols. Details will be added to the revised paper. A full description of the MARGA set-up and operation at the Auchencorth site is available in Twigg et al. (2015). The processed and ratified MARGA data are publicly available online at https://uk-air.defra.gov.uk/data/data_selector from which concentrations of any of the species measured by the MARGA can be selected. We will add the references and online resources to the paper.

References

Erisman, J. W. and Wyers, G. P.: Continuous measurements of surface exchange of SO2 and NH3; Implications for their possible interaction in the deposition process, Atmos. Environ. Part A. Gen. Top., 27(13), 1937–1949, doi:10.1016/0960-1686(93)90266-2, 1993.

Flechard, C. R. and Fowler, D.: Atmospheric ammonia at a moorland site. II: Long-term surface-atmosphere micrometeorological flux measurements, Q. J. R. Meteorol. Soc., 124(547), 759–791, doi:10.1002/qj.49712454706, 1998.

Milford, C., Theobald, M. R., Nemitz, E. and Sutton, M. A: Dynamics of ammonia exchange in response to cutting and fertilising in an intensively-managed grassland, Water, Air Soil Pollut., 1, 167–176 [online] Available from: http://link.springer.com/article/10.1023/A:1013142802662, 2001.

Milford, C., Theobald, M. R., Nemitz, E., Hargreaves, K. J., Horvath, L., Raso, J., Dämmgen, U., Neftel, A., Jones, S. K., Hensen, A., Loubet, B., Cellier, P. and Sutton, M. A.: Ammonia fluxes in relation to cutting and fertilization of an intensively managed grassland derived from an inter-comparison of gradient measurements, Biogeosciences, 6(5), 819–834, doi:10.5194/bg-6-819-2009, 2009.

Neirynck, J., Janssens, I. A., Roskams, P., Quataert, P., Verschelde, P. and Ceulemans, R.: Nitrogen biogeochemistry of a mature Scots pine forest subjected to high nitrogen loads, Biogeochemistry, 91(2–3), 201–222, doi:10.1007/s10533-008-9280-x, 2008.

Nemitz, E., Sutton, M. A., Choularton, T. W., Wyers, P. G. and Otjes, R. P.: Investigations into vertical gradients of the gas-aerosol phase equilibrium NH3(g) - NH4(s, aq) 4 above vegetative surfaces, J. Aerosol Sci., 29(SUPPL.2), S885–S886, doi:10.1016/S0021-8502(98)90625-5, 1998.

Nemitz, E., Sutton, M. A., Schjoerring, J. K., Husted, S. and Paul Wyers, G.: Resistance modelling of ammonia exchange over oilseed rape, Agric. For. Meteorol., 105(4), 405–425, doi:10.1016/S0168-1923(00)00206-9, 2000.

Nemitz, E., Loubet, B., Lehmann, B. E., Cellier, P., Neftel, A., Jones, S. K., Hensen, A., Ihly, B., Tarakanov, S. V and Sutton, M. A.: Turbulence characteristics in grassland canopies and implications for tracer transport, Biogeosciences, 6(8), 1519–1537, doi:10.5194/bg-6-1519-2009, 2009.

Ryder, J., Polcher, J., Peylin, P., Ottlé, C., Chen, Y., Van Gorsel, E., Haverd, V., McGrath, M. J., Naudts, K., Otto, J., Valade, A. and Luyssaert, S.: A multi-layer land surface energy budget model for implicit coupling with global atmospheric simulations, Geosci. Model Dev., 9(1), 223–245, doi:10.5194/gmd-9-223-2016, 2016.

Sutton, M. A., Milford, C., Dragosits, U., Place, C. J., Singles, R. J., Smith, R. I., Pitcairn, C. E. R., Fowler, D., Hill, J., ApSimon, H. M., Ross, C., Hill, R., Jarvis, S. C., Pain, B. F., Phillips, V. C., Harrison, R., Moss, D., Webb, J., Espenhahn, S. E., Lee, D. S., Hornung, M., Ullyett, J., Bull, K. R., Emmett, B. A., Lowe, J. and Wyers, G. P.: Dispersion, deposition and impacts of atmospheric ammonia: Quantifying local budgets and spatial variability, Environ. Pollut., 102(SUPPL. 1), 349–361, doi:10.1016/S0269-7491(98)80054-7, 1998.

Thomas, R. M., Trebs, I., Otjes, R., Jongejan, P. A. C. C., Ten Brink, H., Phillips, G., Kortner, M., Meixner, F. X. and Nemitz, E.: An automated analyzer to measure surface-atmosphere exchange fluxes of water soluble inorganic aerosol compounds and reactive trace gases, Environ. Sci. Technol., 43(5), 1412–1418, doi:10.1021/es8019403, 2009.

Twigg, M. M., House, E., Thomas, R., Whitehead, J., Phillips, G. J., Famulari, D., Fowler, D., Gallagher, M. W., Cape, J. N., Sutton, M. A. and Nemitz, E.: Surface/atmosphere exchange and chemical interactions of reactive nitrogen compounds above a manured grassland, Agric. For. Meteorol., 151(12), 1488–1503, doi:10.1016/j.agrformet.2011.06.005, 2011.

Twigg, M. M., Di Marco, C. F., Leeson, S., Van Dijk, N., Jones, M. R., Leith, I. D., Morrison, E., Coyle, M., Proost, R., Peeters, A. N. M., Lemon, E., Frelink, T., Braban, C. F., Nemitz, E. and Cape, J. N.: Water soluble aerosols and gases at a UK background site – Part 1:

Controls of PM2.5 and PM10 aerosol composition, Atmos. Chem. Phys., 15(14), 8131–8145, doi:10.5194/acp-15-8131-2015, 2015.

Wesely, M. L., Cook, D. R., Hart, R. L. and Speer, R. E.: Measurements and parameterization of particulate sulfur dry deposition over grass, J. Geophys. Res. Atmos., 90(D1), 2131–2143, doi:10.1029/JD090iD01p02131, 1985.

Wichink Kruit, R. J., van Pul, W. A. J., Otjes, R. P., Hofschreuder, P., Jacobs, A. F. G. and Holtslag, A. A. M.: Ammonia fluxes and derived canopy compensation points over non-fertilized agricultural grassland in The Netherlands using the new gradient ammonia-high accuracy-monitor (GRAHAM), Atmos. Environ., 41(6), 1275–1287, doi:10.1016/j.atmosenv.2006.09.039, 2007.

Wolff, V., Trebs, I., Foken, T. and Meixner, F. X.: Exchange of reactive nitrogen compounds: Concentrations and fluxes of total ammonium and total nitrate above a spruce canopy, Biogeosciences, 7(5), 1729–1744, doi:10.5194/bg-7-1729-2010, 2010.

---

## Author Response (AR1)

acp-2018-603: Surface-atmosphere exchange of water-soluble gases and aerosols above agricultural grassland pre- and post-fertilisation by Ramsay, R. et al.

5

**Response to Anonymous Referee #1**

We thank the reviewer for their time in reviewing our paper and providing comments. Before addressing the numbered points made by this reviewer individually, we would like to respond to the reviewer's overall question "does this manuscript offer a real advancement of the current state of knowledge?" We believe our paper provides novelty and advancement for the following reasons:

Firstly, there is currently a lack of data for simultaneous time-resolved fluxes of trace inorganic gases and associated aerosol counterparts, particularly for the reactive nitrogen species NH3/NH4+ and HNO3/NO3-. Our paper pre-

15 sents flux data for these species over agricultural grassland at hourly resolution for one month at high precision and with appropriate consideration of the uncertainties in the flux values. This dataset also includes fluxes for trace gas and aerosol species during a period of flux divergence post-application of urea fertiliser. By careful consideration of the issues present in analysing fluxes during periods of flux divergence, we present a robust dataset which considers total nitrate and total ammonium fluxes during this period and discusses the changes in flux behaviour

20 post-fertiliser application. Furthermore, it provides strong field evidence of a ground source of both HONO and HNO3 after fertilisation.

Secondly, this paper presents bulk deposition velocities for particulate Cl-, NO3- and SO42-, which are themselves important values for deposition modelling. From observation of these deposition velocities, it was hypothesized

25 there was a link between deposition velocity and the proportion of fine to coarse aerosol. Using the ratio of PM2.5/PM10, as measured by a nearby instrument, we were able to demonstrate this association. While this proportion acts as a proxy measurement for particle size measurements, we believe this is novel evidence for demonstrating a link between enhancement of aerosol deposition velocity and proportion of particles contained in the coarse fraction.

30

Thirdly, we believe we present the first intercomparison data for nitrous acid measurements made by the Gradient of Aerosols and Gases Online Registration (GRAEGOR) and by the Long Path Absorption Photometer (LOPAP), which not only compares concentrations, but also gradients. We also present an intercomparison of ammonia measurements made by the GRAEGOR and by a Quantum Cascade Laser (QCL).

35

We now respond to the individual points raised in the review, with the reviewer's comments presented in blue, italicized font.

 "As illustrated by many of the figures using gradients to determine fluxes is extremely difficult (see the concentration plots in Fig 3 & 4). Indeed direct flux measurements are also very difficult! Thus although the fluxes shown in Figure 7 and Figure 9 are presented without error bars I suspect the error bars are in fact VERY large. This is not a new problem and is certainly not unique to these authors or this study. BUT Figure 10 actually tells an important part of the story as does in Figure 14...that the concentrations are themselves rather uncertain."

45 We agree that it would be helpful to include error bars on the time series for fluxes and will add these to Figures 7 and 9 in the revised manuscript. We have included in this response a revised version of Figure 7 – the time series of trace gas fluxes - which includes error bars as an example. We will also add a summary of median flux error values to Tables 4 and 5.

Figure 7: Time series of hourly trace gas fluxes measured during the Easter Bush campaign. Results smoothed using a 5-point moving point average. The fertilisation period was 08:00 – 09:00 on 13th June, and is highlighted in green. Flux uncertainties for each trace gas are included as error bars.

Figure 10 (which compares measurements of NH3 taken by the GRAEGOR to that by a QCL system) and Figure 14 (similarly comparing measurements of HONO taken by the GRAEGOR to that by a LOPAP) are intercomparison studies between instruments. These comparisons do not reflect the uncertainty of GRAEGOR concentration measurements, only that there exists a difference between measurement techniques that should be accounted for when

- 5 considering measurements of concentration. Crucially, the difference in measurements between two different systems does not directly impact on the error in the concentration gradient of the GRAEGOR, which is the critical part in calculating flux values. The two GRAEGOR detector boxes share the same analytical system and therefore uncertainties in the concentrations at the two heights are not independent, and the error on the gradient is significantly smaller than the combined error between the two concentrations.
- 10

15

It is our view that the flux uncertainties in this study are not "very large" in respect to previously published studies using the GRAEGOR. The median flux error for NH3, for example, was 32%, which is similar to values obtained for measurements made over grassland using the GRAEGOR by Wolff et al. (2010) and Thomas et al. (2009). By inclusion of median flux error values in Tables 2 and 3 of the revised manuscript, we anticipate we can satisfy readers that our flux errors are in the range expected for use of this instrument.

2) "Fundamentally is GRAEGOR 'fit for purpose'? Some basic statistics could be brought into play to consider what fraction of flux periods (of each of the considered species) exceeded the uncertainty bounds FOR each individual measurement."

20

25

The reviewer raises an important point that was considered during flux calculations, but which was not included in the manuscript. As outlined by Thomas et al. (2009), it is possible to calculate the minimum flux that the GRAEGOR can measure for each species, effectively providing a limit of detection for flux measurements. Fluxes presented have been filtered using these values, but discussion of their development, beyond mentioning that fluxes were filtered according to these values was not included, nor were they included in Tables 4 and 5. We will include a

brief discussion of calculating this value in the Methods section of the revised manuscript, which, together with inclusion of minimum detectable flux values for each species in Tables 2 and 3, should resolve this issue.

It is important to emphasize that the capability of the GRAEGOR to measure fluxes is dependent upon its ability to measure concentration differences with sufficient precision. As documented in the manuscript, a side-by-side comparison of the GRAEGOR sample boxes was used to develop a linear regression profile, from which – after correction – the residuals were used to determine the precision of the concentration measurements. As we state in our manuscript: "From the results obtained, it was found that for the gases NH3, HCl, HONO, HNO3 and SO2 that deviation from the 1:1 fit resulted in a precision of measurements <4% (3σ). For the aerosol species Cl-, NO3- and

- 35 SO42°, precision was calculated as <8% (3σ), while for NH4+ was calculated as <9% (3σ)." These precision values are in line with those previously calculated by those using the GRAEGOR to measure fluxes (Thomas et al., 2009; Wolff et al., 2010; Twigg et al., 2011), and we maintain that these are sufficient precision values to resolve the vertical concentration gradients necessary for flux calculations.
- 40 The GRAEGOR shares its principle of operation and many components with other instrumentation that has routinely been used for gradient flux measurements, such as the AMANDA/GRAHAM for NH3 (same denuder but based on selective membrane / conductivity; Erisman and Wyers, 1993; Flechard and Fowler, 1998; Milford et al., 2001, 2009; Wichink Kruit et al., 2007; Neirynck et al., 2008) and the MARGA-based gradient system of Rumsey et al. (2016). Considering the similar architecture of the GRAEGOR to these instruments which have been suctioned by the standard flux and the the theter the CRAEGOR to these instruments which have been suctioned by the standard flux and the standard flux and the the theter the standard flux and the standa
- 45 cessfully used to measure gradient fluxes, the statement that the GRAEGOR would not be fit for purpose is therefore surprising.

Finally, the good agreement between expected and measured deposition velocities for HNO3 and HCI may be taken as independent evidence (though not proof) of the high quality of the measured fluxes.

"The authors describe some efforts at determining uncertainty in concentrations and fluxes but they do not appear to be applied"

As described in responses above, we will resolve issues of clarity surrounding the uncertainty measurements by 5 revising quoted figures in the text so that they include their error values. We will also include error bars in figures where appropriate, and revise Tables 2 and 3 to include further details on error measurements.

"...the description is quiet[sic] vague and associated with statements I find it hard to comprehend; 'Uncertainties for the trace gases and water-soluble aerosols measured calculated by error propagation ranged from 8% - 18%
(3σ) throughout the campaign, varying primarily due to fluctuations in the measured flow rate and analysed concentration of the internal Br standard.' Does this really mean ALL species for ALL hours had an uncertainty of 8-18% of the measured concentration?"

We agree with the reviewer that the wording of this section lacks clarity. The determination of concentration meas-15 urement error results in the error for concentration measurement. Side-by-side measurements measure the error in concentration difference. Error in flux calculations are determined from this error in concentration difference and the error in flux transfer velocity. The error of 8 – 18% in concentration measurements is reasonable in comparison to past campaigns with the GRAEGOR (Thomas et al., 2009; Wolff et al., 2010). We will clarify this issue in the revised manuscript.

' $\sigma u^*$  was estimated at 12% median, which, in combination with  $\sigma_{\Delta c_r}$  was used to calculated  $\sigma F' - I$  can't see uncertainties are presented... 'While most exceedances fall within the uncertainty range of the measurement' How many do not? And why?

3) Addressing point 2) and doing so in a manner that actually uses uncertainties for EACH measurement not for
 the sample as a whole would be useful in contextualizing the flux estimates and allowing the authors to determine
 if the 'good enough' threshold is achieved."

As discussed above, we shall resolve this issue through the inclusion of the necessary values in the text.

35

30 4) "I think Figure 12 is partly a response to particle size but since no data on particle size were provided is it also a story of large measurement uncertainty?"

In our discussion of the investigation into the dependence of measured deposition velocity ( $V_d$ ) on particle size, we repeatedly emphasize that our value for particle size is a proxy (the ratio  $PM_{2.5}/PM_{10}$  as measured by an instrument nearby). As stated in the text – "Although measurements of particle size were not made during this campaign, measurements of aerosol species (including Cl- and SO42-) in the  $PM_{2.5}$  and  $PM_{10}$  size fractions were taken by a two-channel Monitor for Aerosols and Gases in Ambient Air (MARGA, Applikon B.V, The Netherlands) instrument

- located at Auchencorth Moss, 12 km south west of Easter Bush...As proxy for a particle size measurement, the proportion of PM2.5 to PM10 was used, with a lower proportion of PM2.5 indicating a greater proportion of coarse aerosol, and a corresponding larger deposition velocity based on process-orientated modelling". Particle size
- measurements were not available. As summarised at the start of this response document, the proxy measurement was used to investigate a hypothesis that was developed from observations of aerosol Vd values. Our use of such a proxy measurement is not related to measurement uncertainty. Figure 12 shows strong and statistically significant relationships. The scatter indicated may reflect measurement uncertainty, but could equally reflect limitations in a
- 45 concentration ratio from a nearby site to describe the full size-distribution at our measurement site, the additional effect of atmospheric stability on Vd/u- (e.g. Wesely et al., 1985) or a number of additional processes (e.g. surface wetness). We anticipate that the reviewer's concern will be resolved by further clarifying the proxy nature of the aerosol size measurement in our manuscript.

**Commented [RR1]:** A revised statement from the original response, which directly clarifies the calculation of concentration measurement error and flux error. 5) "The manuscript title implies a focus on fluxes ("Surface-atmosphere exchange of water-soluble gases and aerosols above agricultural grassland pre- and post-fertilisation") why are so many of the figures and so much of the text about concentrations and/or the ion balance in the aerosols?

- 5 We believe it is necessary to include figures and text discussing concentrations as a precursor to discussion of fluxes. It is also important to present these findings for the discussions relating to (i) the deposition velocities (which makes references to elevated periods of CI concentrations that are visually apparent in Figure 3), (ii) the HONO fluxes (the presence of HONO concentrations above the detection limit suggests a day time source for HONO; this is then linked to discussion of HONO fluxes), and (iii) the instrument intercomparison studies (which compare concentrations). 10

The inclusion of a brief discussion of ionic balance, with an accompanying figure, was necessary to discuss the development of the hypothesis of aerosol bulk deposition velocities being enhanced by particles in the coarse fraction. As mentioned in the text - "[the ionic balance study]...suggests a deficit of NH4+, su

---

## Author Response (AR2)

**acp-2018-603: Surface–atmosphere exchange of inorganic water-soluble gases and associated ions in bulk aerosol above agricultural grassland pre- and post-fertilisation**
**by Ramsay, R. et al.**

**Response to Anonymous Referee #4**

We thank the reviewer for their time in reviewing our paper. The reviewer has provided an additional comment in their report, and we shall address their concern in this response.

*"My only additional comments regards the concentration footprint: flux and concentration footprints are generally different, with the concentration ones having a wider area compared with the flux ones. Is the area around the sampling point homogeneous enough for discussing fluxes and concentrations "hand-in-hand" (also considering flux divergencies)?"*

We struggle to fully understand the point the reviewer is making and consequently to address their concern. It is correct that concentration and flux footprints are very different, with the flux footprint extending to typically 100× the measurement height (depending on atmospheric stability) and the concentration footprint being much larger. We do discuss both concentrations and fluxes in the paper, but at no point state that the concentrations are only affected by the field itself. By contrast we did confirm through footprint estimation that the reported fluxes were dominated by the field itself. Otherwise a flux measurement by the aerodynamic gradient technique would indeed not be meaningful. For this assessment it is important to understand that for gradient flux measurements the flux footprint describes the footprint of the *concentration difference* between the two heights rather than the footprint of the concentration measured at any single height itself.

Flux divergence on the other hand is caused by the local interaction between vertical transport and chemistry, depending on the relative time-scales of the two processes.

Whilst the footprints are different, the exchange mechanisms studied above this particular field will obviously apply more widely and therefore affect regional concentrations. For example, the elevated deposition velocity for the coarse aerosol fraction will also apply to the wider landscape and result in a decreased lifetime and transport distance of this aerosol fraction during transport to the site.

In summary, we do not see any conflict in discussing both concentrations and fluxes in the present manuscript and cannot see where we may have confused the reviewer.

**Response to Co-Editor**

We thank the co-editor for overseeing the review process for this paper.

We have noted the co-editor's concern over the length and coherence of the introduction, and we have therefore taken steps which resolve this. With regards to length, we have shortened the introduction by 231 words, from 1792 to 1561 words, by omitting supplementary information that was not essential for the overall study justification. We have also improved clarity by rearranging sections of the introduction. The justification for the study is presented in the first paragraph, followed by short introductions to the inorganic trace gases and associated aerosol counterparts measured, with a brief overview of the aerodynamic gradient method and instrumentation.